# Soil net nitrogen mineralisation across global grasslands

A.C. Risch (ID) et al.#

Soil nitrogen mineralisation ($N_{min}$), the conversion of organic into inorganic N, is important for productivity and nutrient cycling. The balance between mineralisation and immobilisation (net $N_{min}$) varies with soil properties and climate. However, because most global-scale assessments of net $N_{min}$ are laboratory-based, its regulation under field-conditions and implications for real-world soil functioning remain uncertain. Here, we explore the drivers of realised (field) and potential (laboratory) soil net $N_{min}$ across 30 grasslands worldwide. We find that realised $N_{min}$ is largely explained by temperature of the wettest quarter, microbial biomass, clay content and bulk density. Potential $N_{min}$ only weakly correlates with realised $N_{min}$, but contributes to explain realised net $N_{min}$ when combined with soil and climatic variables. We provide novel insights of global realised soil net $N_{min}$ and show that potential soil net $N_{min}$ data available in the literature could be parameterised with soil and climate data to better predict realised $N_{min}$.

---

#A full list of authors and their affiliations appears at the end of the paper.

Soil nitrogen (N) availability is one of the most important drivers of ecosystem productivity[1–3] and microbial decomposition[4], and is key in regulating N cycling. During the breakdown and depolymerisation of organic material to monomers and inorganic N, plants and microbes compete for these N resources[5–7]. The net balance of N mineralisation and immobilisation (soil net $N_{min}$) is mediated by soil physico-chemical properties, aboveground and belowground litter input, plant and microbial nutrient demand and climatic factors[5,8–13], and is regarded as a good index of overall soil N availability[5]. Soil net $N_{min}$ usually is greater in well-aerated soils from more humid climates at lower latitudes[14–16], reflecting the controls of soil temperature, moisture and oxygen content over microbial activity. However, climatic conditions also shape soil properties over long time-scales[17,18], so understanding the impact of climate and soil properties for soil net $N_{min}$ is crucial to achieve robust estimates of soil N availability, and ultimately, productivity in terrestrial ecosystems.

Soil net $N_{min}$ is commonly estimated either in the field or laboratory. Field measures represent realised soil net $N_{min}$ constrained by site macro-climatic and micro-climatic conditions (Fig. 1) and are typically collected at local to regional scales[12]. In contrast, studies at the continental[15,16] or global scale[14,19] generally rely on laboratory incubations that estimate potential soil net $N_{min}$. Laboratory incubations use homogenised soil samples incubated under optimised and controlled temperature and

moisture conditions to allow the comparison of samples collected from different locations in a standardised way. Yet, homogenisation disrupts the soil structure and removes plant residues, which may affect these estimates[20,21]. Laboratory measures also fail to account for soil micro-climatic differences found under field conditions. Although potential may in some cases effectively predict realised soil net $N_{min}$, we do not know whether, or under what conditions this is the case. Successfully identifying the environmental drivers connecting the two indices could greatly enhance the use of potential soil net $N_{min}$ data to model and predict global patterns in realised soil net $N_{min}$. This would facilitate our understanding of how global soil N availability may respond to future global anthropogenic influences such as climate change or eutrophication[22,23].

We conducted coordinated measurements of realised and potential soil net $N_{min}$, and assessed soil properties and climatic variables across 30 grasslands on six continents that span a globally relevant range of climatic and edaphic conditions (Fig. 2; Supplementary Tables 1 and 2). We focused on grassland ecosystems because they cover approximately one-third of the Earth's terrestrial landscape[24,25], are threatened by global change[22,23,26], and provide important ecosystem services intricately linked with $N_{min}$. Notably, they store 20–30% of the terrestrial C, mostly in the soil[24,25,27]. Here, we describe the global spatial patterns in realised and potential soil net $N_{min}$ and the relationship between them. We then explore the key drivers of each soil net $N_{min}$ index

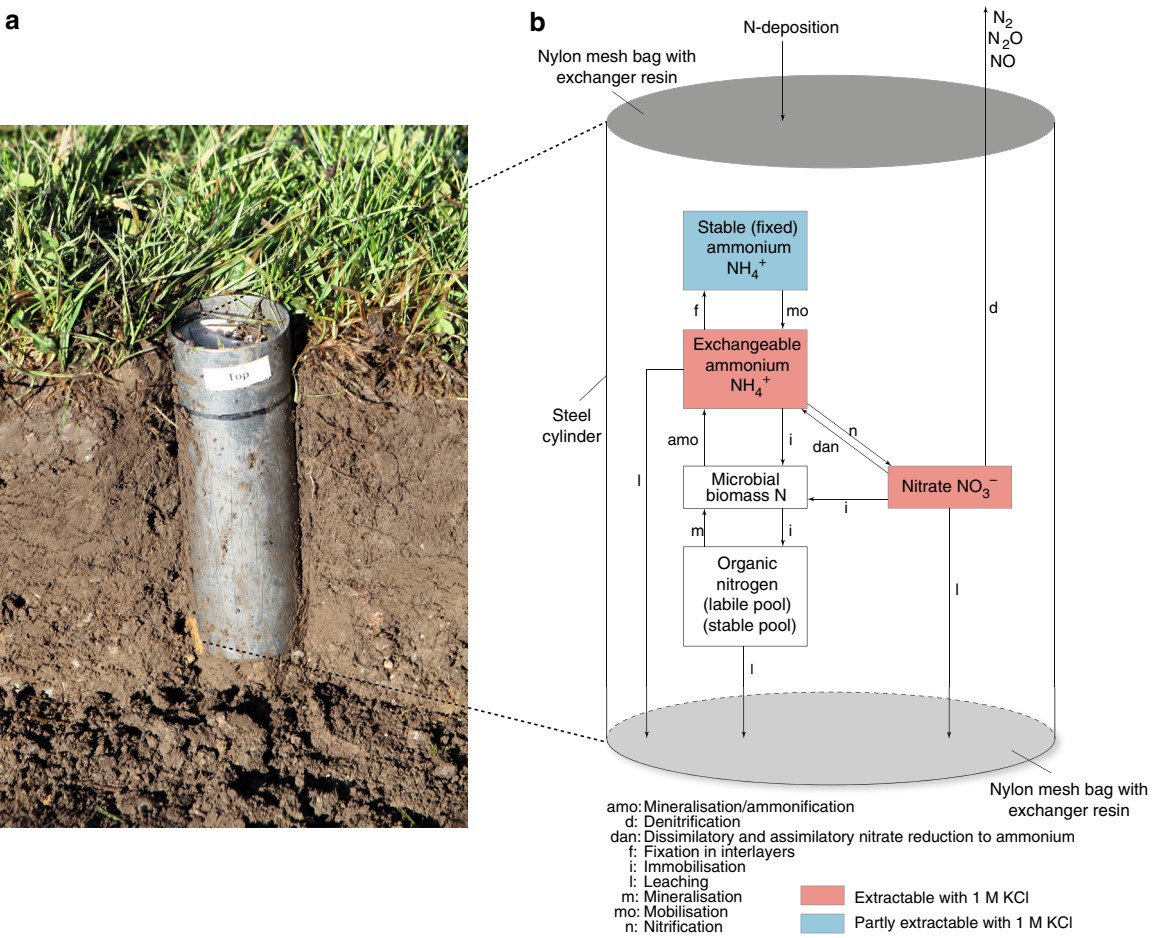

**Fig. 1** Realised soil net N mineralisation. **a** Photo of a cylinder used during field measurements. A mesh-bag filled with ion-exchange resin is visible at the top of the cylinder. **b** Schematic N mineralisation processes as found within our cylinders. An exchange resin bag on top captured atmospheric N deposition/N in run-off, another resin bag at the bottom of the cylinder captured N leaching from the soil column. Details on calculation of soil net N mineralisation based on the variables measured are given in the "Methods" section

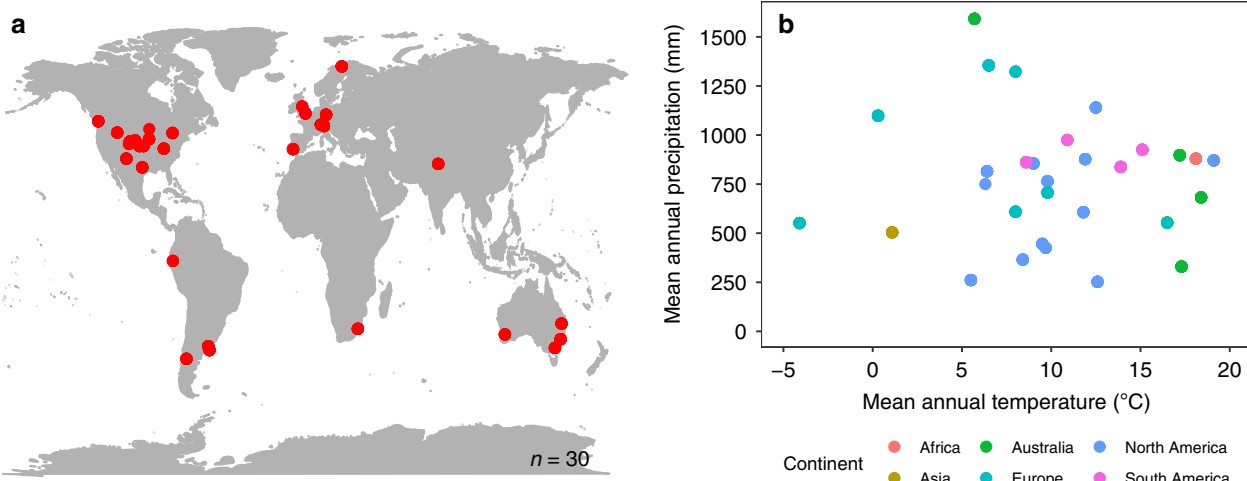

**Fig. 2** Geographic and climatic distribution of experimental sites. **a** Location of the 30 NutNet sites where the field experiment was conducted and soil samples were collected for laboratory analyses. **b** The 30 study sites represent a wide range of mean annual temperature (MAT) and mean annual precipitation (MAP). Our sites also cover a wide range of soil edaphic conditions as described in the main text and shown in Supplementary Table 2. Source data are provided in the source data file

separately. Finally, we use structural equation modelling (SEM) to build a system-level understanding of how these specific climate and soil variables together with potential soil net $N_{min}$ could be used to predict realised soil net $N_{min}$. This final step provides a basis upon which extensively available data on potential soil net $N_{min}$ could be leveraged to improve predictions of realised soil net $N_{min}$ at a global scale.

First, we expect opposite global spatial patterns in realised and potential soil net $N_{min}$: Namely, decreasing realised soil net $N_{min}$ with increasing distance to the equator due to colder growing season temperatures and therefore lower mineralisation. Concurrently, we expect increasing potential soil net $N_{min}$ with increasing distance to the equator, because higher amounts of labile organic material accumulated under colder conditions are mineralised when incubated under standardised conditions in the laboratory. Second, climate variables and the presence of plant residues should predict realised soil net $N_{min}$ through their effects on soil properties and the activity of soil biota[11–13]. In contrast, soil chemical properties, soil texture and the activity/abundance of soil biota may be more important for potential soil net $N_{min}$ than climatic variables[11–13]. Nonetheless, as climatic conditions impact soil formation[17,18], we expected that climate provides additional predictive information to explain potential soil net $N_{min}$. Third, we expect that realised soil net $N_{min}$ can be estimated by combining key environmental drivers with potential soil net $N_{min}$.

All 30 sites (Fig. 2, Supplementary Tables 1 and 2) are part of the Nutrient Network globally distributed experiment (NutNet [https://nutnet.umn.edu/])[28]. We incubated one soil sample per plot in the field to assess realised soil net $N_{min}$ (Fig. 1)[29] and collected additional samples to measure potential soil net $N_{min}$ and soil properties, i.e., water holding capacity, bulk density, C and N content, texture, pH, pore space, microbial biomass, and archaeal (AOA) and bacterial (AOB) ammonia oxidiser abundance using identical materials and methods at each site. The field incubation averaged 42 days (range 36–57) and ended at peak plant biomass at each site. Soil moisture was kept at 60% of the field capacity of each sample and 20 °C for the 42-day laboratory incubations. Climate data for each site were obtained from global data sets[30].

We dropped correlated variables prior to analyses[31], calculated the correlation between realised and potential soil net $N_{min}$ and then explored the global spatial patterns in realised and potential soil net $N_{min}$ with linear mixed effects models (LMMs). Then, we

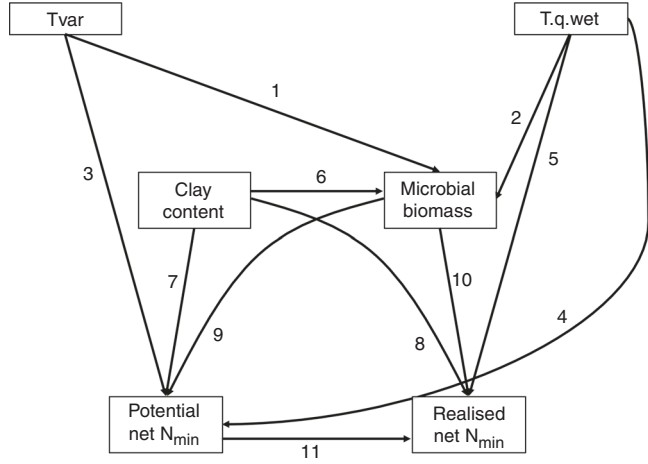

**Fig. 3** Conceptual model on the expected causal relationships between environmental variables, soil properties and potential soil net $N_{min}$ to estimate realised soil net $N_{min}$. The conceptual model is based on hypotheses derived from the literature and our linear mixed effects model results (see Table 1 for hypotheses and references). Tvar = temperature variability, T.q.wet = temperature of the wettest quarter

used LMMs and multilevel inference[32] to determine the drivers of realised and potential soil net $N_{min}$. Based on the LMMs and existing literature, we developed a conceptual model of causal relationships among environmental drivers, potential soil net $N_{min}$ and realised soil net $N_{min}$ (Fig. 3 and Table 1)[33].

Here, we find that realised correlates only weakly with potential soil net $N_{min}$, and that different environmental parameters drive the two measures. However, potential soil net $N_{min}$ contributes to explain realised $N_{min}$ when combined with soil and climatic variables. We provide new insights for realised soil net $N_{min}$ and show how potential soil net $N_{min}$ data could be parameterised to better predict realised $N_{min}$ in global grasslands.

## Results and discussion
**Global patterns and drivers of soil net N mineralisation**. Across our 30 grassland sites, realised and potential soil net $N_{min}$ were weakly positively correlated ($r = 0.21$, $p = 0.052$, $df = 83$,

**Table 1 Potential relationships between variables derived from the literature and the linear mixed effects model results (see Table 2)**

| Path # | Pathway | Proposed relation |
|---|---|---|
| 1, 2 | Temperature variability or temperature of the wettest quarter → microbial biomass | The variability in temperature alone or in combination with precipitation affect microbial biomass across large scales[19,60,61] |
| 3, 4 | Temperature variability or temperature of the wettest quarter → potential $N_{min}$ | The best LMMs revealed that temperature variability had a positive, and temperature of the wettest quarter a negative effect on potential soil net $N_{min}$ (Table 2). These two direct effects likely stand for surrogates of missing direct drivers influenced by climate parameters that we could not identify in this current study. Thus, temperature variability and temperature of the wettest quarter represent a legacy effect of long-term climatic properties on soils in a global context. |
| 5 | Temperature of the wettest quarter → realised $N_{min}$ | The top LMMs in this study revealed that temperature of the wettest quarter has a positive effect on realised soil net $N_{min}$ (Table 2) across global grasslands. |
| 6 | Clay content → microbial biomass | Clay content positively affects microbial biomass on a global scale[14,16] |
| 7, 8 | Clay content → potential and realised $N_{min}$ | Clay content positively affects potential and realised $N_{min}$[14,16] |
| 9, 10 | Microbial biomass → potential and realised $N_{min}$ | Microbial biomass has a positive effect on soil mineralisation rates $N_{min}$[19] |
| 11 | Potential $N_{min}$ → realised $N_{min}$ | The goal of this study was to determine whether knowledge about the potential of a soil to mineralise N can predict realised $N_{min}$, but we only found a weak correlation between potential and realised $N_{min}$ (see main text, Supplementary Figure 1). However, the potential of soil microbes to mineralise N may be revealed when we include several climatic as well as soil biotic and abiotic predictors simultaneously into a model as done here. Thus, higher potential $N_{min}$ measured in the laboratory should result in higher realised $N_{min}$ in the field across global grasslands. |

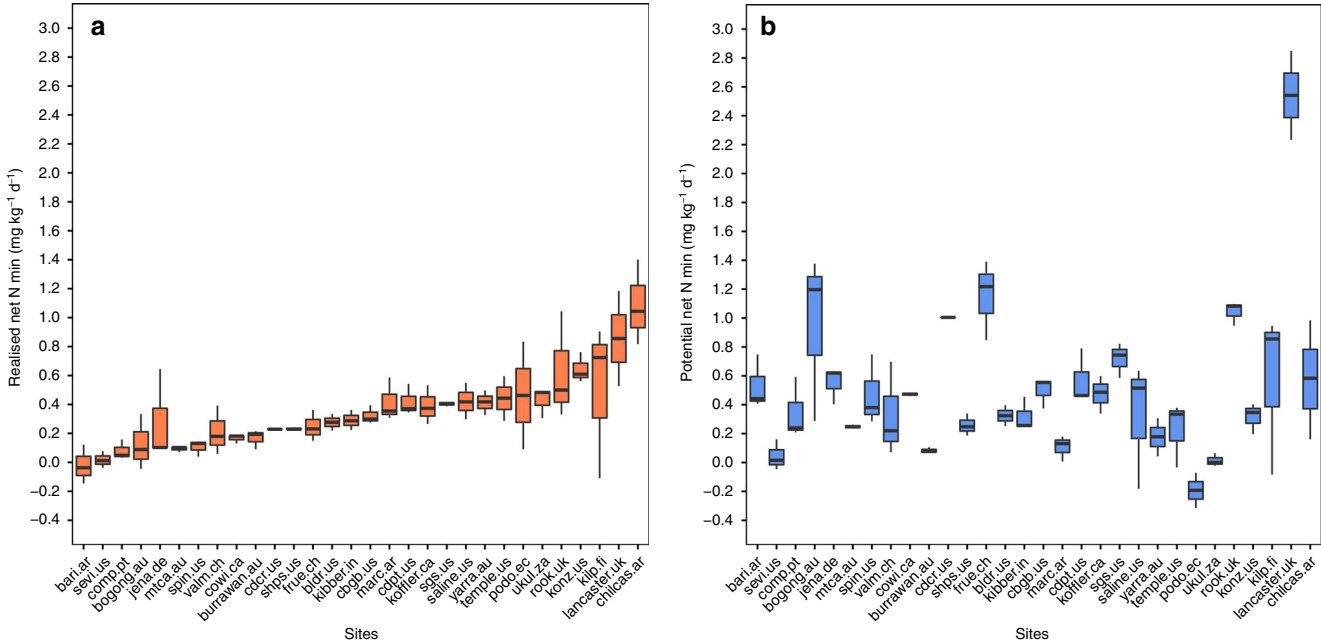

**Fig. 4** Global patterns in realised and potential soil net N mineralisation (soil net $N_{min}$). **a** Realised soil net $N_{min}$ at 30 NutNet sites ordered according to increasing realised soil net $N_{min}$. **b** Potential soil net $N_{min}$ at the 30 NutNet sites. Order of sites according to **a**. The box represents the median (50th percentile), 25th and 75th percentile of the data for each site. The whiskers represent 1.5 times the inter-quartile range. Source data are provided in the source data file

Supplementary Fig. 1). Individual values of realised soil net $N_{min}$ ranged from −0.14 to 1.40 mg N kg soil⁻¹ day⁻¹, and from −0.31 to 2.85 mg N kg soil⁻¹ day⁻¹ for potential soil net $N_{min}$ (Fig. 4a, b). Among sites, average realised soil net $N_{min}$ ranged from −0.02 mg N kg soil⁻¹ day⁻¹ at Bariloche, Argentina (bari.ar) to 1.09 mg N kg soil⁻¹ day⁻¹ at Las Chilcas, Argentina (chilcas.ar), whereas average potential soil net $N_{min}$ ranged from −0.19 mg N kg soil⁻¹ day⁻¹ at Podocarpus, Ecuador (podo.ec) to 2.54 mg N kg soil⁻¹ day⁻¹ at Lancaster, UK (lancaster.uk, Fig. 4a, b). Within-site range of realised soil net $N_{min}$ was lowest at Sheep Station, UT, USA (shps.us; $\Delta = 0.02$ mg N kg soil⁻¹ day⁻¹) and highest at Kilpisjärvi, Finland (kilp.fi; $\Delta = 1.01$ mg N kg soil⁻¹

day⁻¹); for potential soil net $N_{min}$, the range was lowest at Mt. Caroline, Australia (mtca.au; $\Delta = 0.03$ mg N kg soil⁻¹ day⁻¹) and highest at Bogong, Australia (bogong.au; $\Delta = 1.08$ mg N kg soil⁻¹ day⁻¹; Fig. 4a, b).

Contrary to our predictions, realised soil net $N_{min}$ was not related to distance to the equator ($F_{1,28} = 0.057$, $p = 0.81$, Supplementary Fig. 2A), even when we corrected the values for growing season length for each site (Supplementary Fig. 3A, B). However, potential soil net $N_{min}$ significantly increased with increasing distance ($F_{1,28} = 22.86$, $p < 0.001$, Supplementary Fig. 2B), which supports our hypothesis but contrasts with meta-analyses[14,15] reporting decreased potential soil net $N_{min}$

**Table 2 Model selection results for realised soil net $N_{min}$ and potential soil net $N_{min}$ starting with the full model including all explanatory variables (Supplementary Table 5) followed by multi-model inference to select the simplest models that explained the most variation in realised and potential soil net $N_{min}$**

| Top models | Exp. vars incl. | Estimate | SE | p | df | AICc |
|---|---|---|---|---|---|---|
| *Realised soil net $N_{min}$* | | | | | | |
| Model 1 | Intercept | 0.521 | 0.037 | <0.001 | 5 | 23.44 |
| | T.q.wet | 0.106 | 0.038 | 0.01 | | |
| | Microbial biomass | 0.142 | 0.037 | <0.001 | | |
| Model 2 | Intercept | 0.520 | 0.033 | <0.001 | 4 | 23.50 |
| | Microbial biomass | 0.125 | 0.039 | 0.002 | | |
| Model 3 | Intercept | 0.521 | 0.041 | <0.001 | 4 | 24.04 |
| | Clay content | 0.121 | 0.039 | 0.003 | | |
| Model 4 | Intercept | 0.518 | 0.036 | <0.001 | 5 | 24.73 |
| | T.q.wet | 0.122 | 0.039 | 0.004 | | |
| | Bulk density | −0.133 | 0.036 | <0.001 | | |
| *Potential soil net $N_{min}$* | | | | | | |
| Model 5 | Intercept | 0.587 | 0.042 | <0.001 | 6 | 68.64 |
| | AOB | 0.123 | 0.039 | 0.003 | | |
| | T.q.wet | −0.194 | 0.045 | <0.001 | | |
| | Tvar | 0.125 | 0.045 | 0.010 | | |
| Model 6 | Intercept | 0.587 | 0.046 | <0.001 | 5 | 69.29 |
| | AOB | 0.134 | 0.041 | 0.002 | | |
| | T.q.wet | −0.149 | 0.046 | 0.003 | | |
| Model 7 | Intercept | 0.592 | 0.064 | <0.001 | 4 | 69.45 |
| | Microbial biomass | 0.163 | 0.059 | 0.007 | | |
| Model 8 | Intercept | 0.589 | 0.052 | <0.001 | 5 | 69.47 |
| | T.q.wet | −0.206 | 0.056 | 0.001 | | |
| | Tvar | 0.152 | 0.054 | 0.009 | | |
| Model 9 | Intercept | 0.590 | 0.057 | <0.001 | 4 | 70.58 |
| | T.q.wet | −0.150 | 0.058 | 0.015 | | |

Model selection criteria were set at delta AICc < 2 due to our small sample size. All results are based on linear mixed effects models with site identity as a random factor. Exp. vars. incl. = All explanatory variables included in the respective model, Estimate = parameter estimate, SE = parameter estimate standard error, p = p-value related to each variable, df = degrees of freedom of the component model, AICc = corrected Akaike's information criterion, T.q.wet = temperature of the wettest quarter, AOB = ammonia oxidising bacteria, Tvar = temperature seasonality. The total number of observations in all models = 85, the total number of sites in all models = 30

microbes. The greater explanatory power of temperature of the wettest quarter, as opposed to MAT or MAP, shows that annual averages may be less useful for predicting ecosystem processes than more temporally specific climatic variables[30,34]; higher temperatures may only promote soil biological activity if soil moisture levels are sufficiently high[35]. Our results also suggest that sites with higher soil clay content and lower soil bulk density likely featured more conducive soil micro-climatic conditions for soil microbes to thrive and allowed for higher realised soil net $N_{min}$. In contrast to microbial biomass, soil clay content and soil organic C, soil bulk density is usually not considered a key predictor of soil N mineralisation[11,12]. Here, we show that bulk density as a measure of favourable soil structure improved predictions of realised soil net $N_{min}$. Future soil net $N_{min}$ studies and simulation studies for soil N cycling may benefit from including bulk density.

Potential soil net $N_{min}$ was higher when more AOB were present at the start of the incubation (Models 5, 6, Table 2). However, there is no mechanistic link between AOB and potential soil net $N_{min}$, because AOB only transform ammonium to nitrate, but do not drive net production of total inorganic nitrogen in the soil. Yet, AOB abundance was positively correlated with potential net nitrification (Supplementary Fig. 4), which is similar to previous findings[36–39]. Further, potential soil net $N_{min}$ was positively influenced by temperature seasonality and was negatively affected by temperature of the wettest quarter (Models 5, 6, 8, 9, Table 2). The same models were selected when we replaced soil microbial biomass with organic C (Models 4–7, Supplementary Table 3). However, higher potential soil net $N_{min}$ was also explained by higher soil microbial biomass alone (Model 7; Table 2) and microbial biomass was positively correlated with potential soil net nitrification (Supplementary Fig. 5). Our results agree with findings of a recent meta-analysis that identified soil microbial biomass as an important driver of potential soil net $N_{min}$[19]. In addition, the effect of microbial biomass on potential soil net $N_{min}$ indicates that a quantitatively improved understanding of the soil microbial community could likely improve soil biogeochemical models[40,41]. In addition, the selection of the two climate variables rather than the expected individual soil physical and chemical variables as predictors of potential soil net $N_{min}$ suggests that there is a long-term legacy effect of climate on these grasslands that we were not able to capture with the soil physico-chemical variables that we measured[17,18].

Soil C:N ratio is often regarded as an important predictor of soil net $N_{min}$ as it determines the transition from net N immobilisation to net N mineralisation[16,42]. However, soil C:N was not important in our study as all but one (temple.us) of our grassland soils had C:N ratios below the critical threshold of 20. Also contrary to our expectations, realised and potential soil net $N_{min}$ were both partially constrained by climatic variables. Interestingly, temperature of the wettest quarter positively influenced realised soil net $N_{min}$, but had a negative effect on potential $N_{min}$ (Table 2, also see Fig. 5). This pattern suggests that N mineralisation may 'acclimatise' along climate gradients[43,44]: greater mineralisation occurring in a warmer and wetter climate may lead to the depletion of easily available organic N pools compared to soils from cooler climates. When incubated under constant temperature in the laboratory, mineralisation rates from substrate-depleted soils from warmer climates were less than those from soils of cold regions where labile organic N has accumulated for centuries[43,44]. Alternatively, physical disturbance and disruption of the soil structure caused by sieving and sample homogenisation may have more profoundly affected the samples from warmer and wetter climates. Finally, it could be that the soil microbes did not perform as well as expected in the laboratory because the 20 °C incubation temperature was considerably lower

with increasing distance to the equator. However, the meta-analyses may not be comparable to our more closely controlled study from a single vegetation type, because they included a wide range of data from different land-use and vegetation types (croplands, wetlands, forests, shrublands, grasslands) and incubation conditions (duration, temperature, and soil moisture).

The variation in realised soil net $N_{min}$ across our 30 grassland sites was jointly explained by positive effects of temperature of the wettest quarter and microbial biomass (Model 1; Table 2); by microbial biomass alone (Model 2, Table 2), clay alone (Models 3, Table 2) or by a positive effect of temperature of the wettest quarter combined with a negative effect of soil bulk density (Model 4; Table 2). Many studies consider soil organic C as one of the main drivers of soil net $N_{min}$. In our study, soil organic C was highly correlated with soil microbial biomass ($r = 0.85$). When replacing microbial biomass with soil organic C the model selection process yielded similar results (Models 1–3; Supplementary Table 3). Together our findings suggest that with higher temperature of the wettest quarter and more microbial biomass (or soil organic C), more organic matter was mineralised by soil

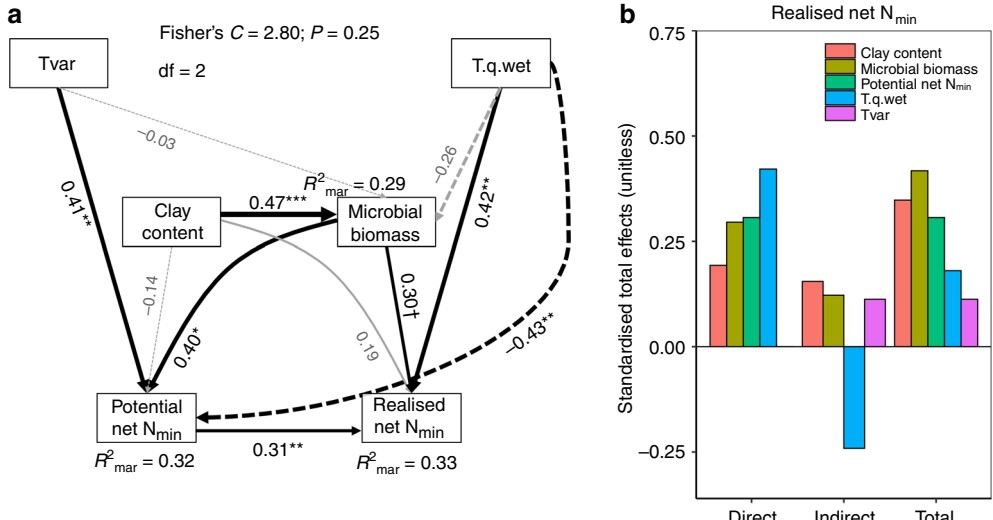

**Fig. 5** Global drivers of realised soil net N mineralisation (soil net $N_{min}$). **a** Structural equation modelling diagram representing connections between climatic conditions, soil physical, chemical and biological properties found to influence realised and potential soil net N mineralisation. The width of the connections represents estimates of the standardised path coefficients, with solid lines representing a positive relationship and dashed lines a negative relationship. Significant connections and $R^2$ are shown in black, non-significant ones in light-grey. **b** Standardised total, direct and indirect effects of variables associated with realised soil net N min. †$p < 0.1$, *$p < 0.05$, **$p < 0.01$, ***$p < 0.001$. Clay content = soil clay content, Microbial biomass = soil microbial biomass, Tvar = temperature seasonality, T.q.wet = temperature of the wettest quarter. The total number of observations = 85, the total number of sites = 30

than the field temperatures during peak growing season at the warmer and wetter sites. Thus, the potentials we measured may not have represented full potential mineralisation for these sites.

**Estimating soil net N mineralisation in grasslands worldwide.** By combining the identified main drivers from the LMMs and potential net $N_{min}$ (Figs. 3 and 5, Table 1), we produced a SEM model that explained 33% (marginal $R^2$) of the variation in realised soil net $N_{min}$ across these grasslands (Fig. 5a). This is similar to the explained variability in potential soil net $N_{min}$ measured in other studies[14,16,19]. The model revealed a new system-level understanding of the controls on global-scale patterns in realised net $N_{min}$ by showing that temperature of the wettest quarter, soil microbial biomass, and potential soil net $N_{min}$ (positive effects) can be directly related to realised soil net $N_{min}$. Soils with higher clay content, which have higher soil microbial biomass, have higher potential soil net $N_{min}$, altogether having a positive effect on realised soil net $N_{min}$ (Fig. 5a, b). The negative effect of temperature of the wettest quarter and the positive effect of temperature variability represent a legacy effect of climate on soil properties that affect potential soil net $N_{min}$. Again, our findings were very similar when we substituted soil microbial biomass with soil organic C (Supplementary Fig. 6).

This study is the first to directly and simultaneously compare realised and potential soil net $N_{min}$ across a globally relevant range of biotic and climatic conditions in grasslands. The two indices were only weakly related across these grasslands, highlighting the uncertainty in using laboratory measurements of soil net $N_{min}$ to predict rates actually occurring in the field. By combining potential soil net $N_{min}$ with specific climate and soil property data, we produced more robust estimations of realised soil net $N_{min}$ across our global set of grasslands. Thus, our results provide a first insight into how potential soil net $N_{min}$ data that is widely available in the literature could be leveraged to learn more about large-scale N mineralisation processes under field conditions. Accurately quantifying realised N mineralisation is crucial for estimating the role of increasing reactive N in ecosystem functioning. Mis-estimation of these processes could lead to

errors in predicting how N limitation affects ecosystem functioning. Given the global extent of grasslands[24,25] this could, in turn, substantially affect our predictions of global change-driven impacts on C cycling[45]. Overall, our findings suggest that management activities that alter soil compaction, nutrient content, or microbial community function may interact with future changes in temperature and precipitation regimes to severely impact the amount of N that is mineralised in grassland soils[22,46].

## Methods
**Study sites and experimental design.** The 30 study sites are part of the Nutrient Network Global Research Cooperative (NutNet [https://nutnet.umn.edu/]; Fig. 2, Supplementary Table 1 and 2). At each site, the effects of nutrient addition and herbivore exclusion treatments are examined via a random-block design[28]. This block design is replicated three times at the majority of the sites. For four sites, we only had data from one (1 site) and two blocks, respectively. This study is restricted to data collected from the untreated control plots (n = 85). Each 5 m × 5 m plot is divided into four 2.5 m × 2.5 m subplots. Each subplot is further divided into four 1 m × 1 m square sampling plots, one of which is set aside for soil sampling[28]. Plots are separated by at least 1 m wide walkways. Mean annual temperature of our sites ranged from −4 to 19 °C, mean annual precipitation from 252 to 1592 mm, and elevations from 6 to 4241 m above sea level (Fig. 2, Supplementary Table 1). Soil organic C varied from 0.32% to 22.30%, soil total N from 0.03% to 1.25% and the soil C:N ratio from 9.07 to 23.64 across our 30 sites. Also soil clay content (3.0–53.3%) and soil pH (3.25–7.71) spanned a large gradient across the 30 sites (Supplementary Table 2). Thus, our 30 sites cover a wide range of grasslands globally that are typical for the respective region (Fig. 2, Supplementary Tables 1 and 2).

**Soil net N mineralisation and soil properties.** Each site received an identical package shipped from the Swiss Federal Institute for Forest, Snow and Landscape Research (WSL) with material to be used for sampling and on-site incubations (steel cores and rings, resin bags, caps, gloves, etc.). For the field incubation, we followed the protocol by Risch et al.[29]. Briefly, at randomised locations in each plot we clipped the vegetation and then we drove a 5 × 15 cm (diameter × depth) steel cylinder 13.5 cm deep into the soil so that 1.5 cm on top of the cylinder remained empty. To capture incoming N from run-off and/or deposition, we placed a polyester mesh bag (mesh-size 250 μm) filled with 13.2 ± 0.9 g of acidic and alkaline exchanger resin (1:1 mixture; ion-exchanger I KA/ion-exchanger III AA, Merck AG, Darmstadt) into the upper 1.5 cm space of the cylinder. The bag was fixed in place with a metal Seeger ring (Bruetsch-Rüegger Holding, Urdorf, Switzerland). Thereafter, we removed 1.5 cm soil at the bottom of the cylinder and placed another resin bag into the cylinder to capture N leached from the soil column. We

made sure that the exchange resin was saturated with $H^+$ and $Cl^-$ prior to filling the bags by stirring the mixture for 1 h in HCl 1.2 M and then rinsing it with demineralised water until the electrical conductivity of the water reached 5 µS/cm. The cylinders including the soil core and the resin bags were then re-inserted into the soil, at the same location where the sample was collected, flushed with the soil surface, and incubated for 42 days (range 36–57, see also Fig. 1a). Each site coordinator chose the timing of incubation so that it started 6 weeks prior to peak plant biomass production. All the incubations were completed between February 2015 and January 2016. At the end of the incubation, the cylinders were re-collected and immediately put in a cool box for transport to the home institution, where they were immediately packed in an insulated box together with cool packs or blue ice to halt further mineralisation, and overnight-shipped to WSL. Gloves were worn at all times to avoid contamination of the samples. Immediately upon arrival at the laboratory at WSL, the resin bags and a 20 g subsample of sieved soil (4 mm) from the cylinders were separately extracted in a 100 ml PE-bottle with 80 ml 1 M KCl for 1.5 h on an end-over-end shaker and filtered through ashless folded filter paper (DF 5895 150, ALBET LabScience). We measured $NO_3^-$ (color-imetrically[47]) and $NH_4^+$ concentrations (flow injection analysis; FIAS 300, Perkin Elmer) on these filtrates.

At the start of the field incubation, we additionally collected two soil cores of 5 cm diameter and 12 cm depth with a steel core at each sampling plot for potential soil net $N_{min}$, soil chemical and biological analyses (see below). We composited the two samples and then re-used the steel cylinder to collect one additional sample (5 × 12 cm) to assess soil physical properties. This third sample remained within the steel core and both ends were tightly closed with plastic caps. The capped steel cores were then gently packed to avoid further disturbance and together with the composited soil samples overnight-shipped to the laboratory at WSL.

From the composited samples, we extracted an equivalent of 20 g dry soil with KCl as described above and $NO_3^-$ and $NH_4^+$ concentrations were measured. Realised soil net $N_{min}$ was then calculated as the difference between the inorganic N content of samples collected at the end of the incubation (plus N extracted from the bottom resin bag) and the N content at the beginning of the incubation and scaled to represent daily mineralisation rates (mg N kg$^{-1}$soil day$^{-1}$)[29]. Note that our values represent soil net $N_{min}$ for an average period of 42 days prior to peak biomass, i.e., typically during the highest biological activity, and not for the entire year.

A second subsample of the composited sample was used to determine potential soil net $N_{min}$ in the laboratory. For this purpose, we weighed duplicated samples of soil equivalent to 8 g dry soil into 50-ml Falcon tubes. Soil moisture was brought to 60% of the field capacity of each individual plot (see methods for determining field capacity below). The Falcon tubes were tightly closed and incubated at 20 °C for 42 days (6 weeks) in a dark room. Every week the Falcon tubes were opened and ventilated. At the end of the incubation, the soil samples were extracted the same way as described above and $NO_3^-$ and $NH_4^+$ determined. Potential soil net $N_{min}$ was calculated as the difference between the N content before and after the incubation and scaled to represent daily values. We also calculated both realised and potential soil net $N_{min}$ on an area basis correcting for soil depth and bulk density, expressed as kg N ha$^{-1}$ 12 cm$^{-1}$ day$^{-1}$ and compared these values against our soil net $N_{min}$ expressed in mg N kg$^{-1}$ soil day$^{-1}$. The two values correlated well for both realised ($R^2 = 0.899$, $p < 0.001$) and potential ($R^2 = 0.900$, $p < 0.001$) soil net $N_{min}$. For comparability among studies we decided to express our soil net $N_{min}$ indices in mg N kg$^{-1}$soil day$^{-1}$.

A third subsample from the composited sample, was sieved (4 mm) and used to assess soil biological properties. Metagenomic DNA was isolated from 0.25 g of bulk soil by using the Qiagen Power soil DNA isolation kit and the extracted DNA was quantified with Pico Green[48]. The abundances of archaeal (AOA) and bacterial (AOB) ammonia oxidisers were quantified using real-time PCR. Functional marker genes encoding archaeal and bacterial ammonia mono-oxygenase (archaeal amoA, bacterial amoA) were quantified by real-time PCR using primers and thermocycling conditions as previously described for bacterial amoA (AmoA-1F and AmoA-2R)[49] and archaeal amoA (Arch-amoAF and Arch-amoAR)[50]. All quantitative PCR assays were carried out with equal amounts of template DNA (2 ng DNA) in 20 µl reactions using QuantiTect SYBR Green PCR master mix (Qiagen, Hirlen, Germany) on an ABI 7500 (PE Applied Biosystems) real-time PCR instrument[48]. The specificity of the amplification products was confirmed by melting-curve analysis, and the expected sizes of the amplified fragments were checked in a 1.5% agarose gel stained with ethidium bromide. Abundances of bacterial and archaeal amoA genes refer to copy numbers per gram dry soil. Standards for qPCR were generated by serial dilution of stocks containing a known number of plasmids carrying the respective functional gene as an insert[51]. Reaction efficiencies of qPCRs were 95% (±2) for archaeal amoA and 93% (±3) for bacterial amoA. $R^2$ values were 0.99 for all runs.

A fourth subsample of the composite sample was sieved (2 mm) and microbial biomass (µg Cmic g$^{-1}$ soil dry weight) was estimated by measuring the maximal respiratory response to the addition of glucose solution (4 mg glucose per gram soil dry weight dissolved in distilled water; substrate-induced respiration method) on ~5.5 g of soil[52]. The rest of the composited sample was dried at 65 °C for 48 h and ground to pass a 2 mm mesh. A subsample was finely ground and analysed for organic C[53] and for total C and N concentrations (Leco TruSpec Analyser, Leco, St. Joseph, MI, USA). We pretreated subsamples with HCl to remove inorganic C prior to soil organic C determination for soils with pH > 7[53]. Mineral soil pH was

measured potentiometrically in 10 mm $CaCl_2$ (soil:solution ratio = 1:2, equilibration time 30 min).

The intact sample within the capped steel core was used to assess field and water holding capacity, fine fraction bulk density, density of the solid phase, soil porosity, and soil texture. For this purpose, the steel core was weighed (without plastic caps), covered with a nylon mesh (SEFAR NITEX 03-60-32, 60 µm mesh) at the bottom and placed in a water bath to saturate the soil core from the bottom-up. The water-saturated soil core was then weighed and placed on a water saturated silt/sand bed with a suction of 60 hPa (pF 1.8; pendant water column of 60 cm) to drain the soil to field capacity. The time elapsing from saturation to drainage of the soil was dependent on soil texture and ranged from 17 and 135 h. After drainage, we weighed the soil (at field capacity), dried it at 105 °C to constant weight and recorded the dry weight. The difference between these two weights corresponds to field capacity. The difference between the weight of the saturated and dried cylinder corresponds to water holding capacity. To calculate the mass of the soil sample, we also weighed the empty steel cylinder and the dried nylon mesh. The density of the solid phase was determined with pycnometers and the pore space was calculated as $\varphi = 1 - \frac{\rho}{\rho_0}$, where $\varphi$ = pore space, $\rho$ = fine fraction bulk density, $\rho_0$ = density of the solid phase. Soil texture was determined with the pipette method[54]. Of the three variables sand, silt and clay, we retained soil clay content as an explanatory variable. A list of all soil chemical, physical and biological properties measured can be found in Supplementary Table 4.

*Climate data*: We selected the following bioclimatic variables that we expected to be important for explaining variability in soil net $N_{min}$ from WorldClim[30] ([http://www.worldclim.org/current], Supplementary Table 4): mean annual temperature (bioclim.T.ann), mean annual precipitation (bioclim.P.ann), the seasonality of annual temperature (bioclim.Tvar), calculated as the standard error of monthly means × 100, temperature of the warmest quarter (bioclim.T.q.warm), temperature of the wettest quarter (bioclim.T.q.wet), temperature of the driest quarter (bioclim.T.q.dry), and precipitation of the wettest quarter (bioclim.P.q.wet; Supplementary Table 4). To obtain a rough estimate of realised soil net N mineralisation over the course of the entire growing season, we obtained daily temperature, precipitation [ftp://ftp.cdc.noaa.gov/Datasets/] and potential evapotranspiration (PET [https://earlywarning.usgs.gov/fews/datadownloads]) data to calculate the total number of growing days for each site. A growing day was defined to have daily precipitation > 0.5*PET and daily minimum temperature > 0.5 °C[55]. All growing days per site were summed to the total length of the growing season. We then multiplied our realised soil net $N_{min}$ with growing season length to assess whether growing season differences among our sites would help to explain latitudinal differences in realised soil net $N_{min}$.

**Numerical calculations and variable selection**. We examined the distributions of our explanatory variables. Some of them were highly skewed and therefore were log transformed. We centred and scaled all explanatory variables to have a mean of zero and variance of one. We then filtered our variables to avoid collinearity between them. For this purpose, we performed a correlation analysis (Supplementary Fig. 7). If variables were strongly correlated (Pearson's $|r| > 0.70$)[31], we selected the ones that allowed us to minimise the number of variables (Supplementary Fig. 7, Supplementary Table 5). In case of the highly correlated variables soil bulk density, soil total C, soil organic C, soil total N, soil pore space and microbial biomass, we chose a soil physical, chemical and biological variable each for use: First, soil bulk density as it is easy and inexpensive to measure and therefore likely to be more commonly available in other studies, second, microbial biomass as the only soil biological variable of the group, and third, soil organic C as it is most often thought to drive soil net N mineralisation. In summary, temperature of the warmest quarter, temperature of the driest quarter, precipitation of the wettest quarter, soil total C, soil total N, and soil pore space were removed from the dataset (Supplementary Table 5). We also transformed our two response variables realised and potential soil net $N_{min}$ (square root transformation) to account for a highly skewed data distribution ($y_t = \text{sign}(y)*\text{sqrt}|y|$; negative values in the data set meant log transformation was not possible).

**Statistical analyses**. To assess the spatial patterns in soil net $N_{min}$, we used linear-mixed effect models (LMMs) fitted by likelihood maximisation using the R nlme package[56] (version 3.131.1) and lme function (R version 3.4.4; R Foundation for Statistical Computing). Realised or potential soil net $N_{min}$, respectively, was the dependent variable, distance to the equator the fixed factor and site identity a random effect. To determine global drivers of grassland soil net $N_{min}$, we used multi-model inference[32] and LMMs. We separately assessed how our variables explained realised soil net $N_{min}$ and potential soil net $N_{min}$. Site identity was used as a random effect in these models. We first calculated the full models including all predictor variables (Supplementary Table 5) and then used the MuMin package[57] (version 1.42.1) to select the simplest models that explained the most variation based on Akaike's information criterion (AIC; model.avg function). We used the corrected AIC (AICc) to account for our small sample size[32,58] and selected the top models that fell within 2 AICc units (delta AICc < 2). We present all our top models rather than model averages. We calculated all our models using either microbial biomass or soil organic C as these two measures represent a very similar soil feature and were highly correlated ($r = 0.85$). The models with soil microbial

biomass are included in the main text, the ones with soil organic C in the Supplementary Information.

Based on findings from the LMM analyses and the literature we developed an a priori causal conceptual model of relationships among environmental drivers, potential and realised soil net $N_{min}$ to test with SEM using a *d-sep* approach[33,59] (Fig. 3, Table 1). The variables included in this model were the best climatic (temperature of the wettest quarter, temperature seasonality), soil texture (clay content) and soil microbial (microbial biomass) predictor variables from the LMM analyses (Figs. 3, 5a, Table 1). We also calculated the same models using soil organic C instead of microbial biomass (Supplementary Fig. 6). In our LMMs, the temperature of the wettest quarter and/or temperature variability predicted realised and potential soil net $N_{min}$. The direct links between climate properties (Tvar, T.q. wet) and potential soil net $N_{min}$ may represent legacy effects of climate on soil properties that we did not directly measure (see main text). Soil clay content was, in turn, predicted to affect microbial biomass (or soil organic C content), realised and potential net $N_{min}$. As we determined microbial biomass prior to incubating the samples in the laboratory or field, we assume that microbial biomass impacts N process rates and not vice versa. Further, as it was the goal of this study to explore if potential soil net $N_{min}$ in combination with other properties could be used to predict realised soil net $N_{min}$, we added a link from potential to realised soil net $N_{min}$. We tested our conceptual model (Fig. 3, Table 1) following a *d-sep* approach using the piecewiseSEM package (version 2.0.2)[59] in R (3.4.0), in which a set of linear structured equations are evaluated individually. This approach allows us to account for nested experimental designs and also to overcome some of the limitations of standard structural equation models such as small sample sizes[33,59]. We first used the lme function of the nlme package to model response variables, including site as a random factor. Good fit was assumed when Fisher's C values were non-significant ($p > 0.05$). Although the abundance of AOB explained some of the variability in potential soil net $N_{min}$, we did not include this variable in our SEMs as there is no mechanistic basis to rationalise that AOB drives the total accumulation of inorganic nitrogen in soil, only its partitioning between ammonium and nitrate. However, we calculated an SEM including AOB to assess if we can predict realised soil net nitrification using our predictors as well as potential soil net nitrification (Supplementary Fig. 8). While we were able to explain potential soil net nitrification well, the model fits rather poorly for realised soil net nitrification (Supplementary Fig. 8).

**Reporting summary**. Further information on research design is available in the Nature Research Reporting Summary linked to this article.

## Data availability

The data will be available at www.envidat.ch. https://doi.org/10.16904/envidat.87, Source data for Figs. 2, 4, 5, Supplementary Figs. 1–8 can be found in the source data file.

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

## Acknowledgements

We appreciate the many helpful comments from two anonymous reviewers that greatly improved the manuscript. This work was conducted within the Nutrient Network (http://www.nutnet.org) experiment, funded at the site-scale by individual researchers. The soil net Nmin add-on study was funded by two internal competitive WSL grants to A.C.R., B.M., M.Sc., F.H. and S.Z. as well as to B.F., A.C.R. and S.Z. Coordination and data management have been supported by funding from the National Science Foundation Research Coordination Network (NSF-DEB-1042132) to E.T.B. and E.W.S., and from the Long-Term Ecological Research (LTER) programme (NSF-DEB-1234162), and the Institute on the Environment at the University of Minnesota (DG-0001-13). We also thank the Minnesota Supercomputer Institute for hosting project data, and the Institute on the Environment for hosting Network meetings. We are grateful to Roger Köchli and Simon Baumgartner for their help with sample processing and analyses, and to Benjamin R. Fitzpatrick for support with calculating growing season lengths and statistical advise. In addition, A.di V. thanks the Nature Conservancy, Gustavo Iglesias and People from Fortin Chacabuco Ranch for access to the field plots to conduct field work there. L.Y. was supported by Universidad de Buenos Aires and Agencia Nacional de Promocion Cientifica y Tecnologica (PICT 2014-3026), and S.M.P. thanks Georg Wiehl for technical assistance, Denise and Malcolm French for use of Mt. Caroline and the support from the TERN Great Western Woodlands SuperSite. N.E. and J.S. acknowledge support of the German Centre for Integrative Biodiversity Research (iDiv) Halle-Jena-Leipzig funded by the German Research Foundation (FZT 118).

## Author contributions

A.C.R., S.Z., M.Sc., F.H. and B.M. developed the idea. A.C.R., R.O.-H. and B.M. analysed the data and M.Sc., J.L.F., F.H., P.B.A. contributed to data analyses. A.C.R. wrote the paper with input of S.Z., R.O.H., M.Sc., B.F., J.L.F., P.A.F., F.H., E.T.B., E.W.S., W.S.H., J.M.H.K., R.L.McC., A.A.D.B., C.J.S., M.L.S., P.B.A., S.B., L.A.B., J.M.B., C.S.B., M.C.C., S.L.C., P.D., A.di V., A.Eb, N.E., E.E., A.Es, N.H., Y.H., K.P.K., A.S.McD., J.L.M., S.A.P., S.M.P., C.R., M.Sa., J.S., K.L.S., P.M.T., R.V., L.Y. and M.B. SZ., B.F. and J.S. analysed the samples. All authors but S.Z., R.O.-H., B.F., F.H., J.S. and B.M. are NutNet site coordinators, collected and shipped the soil data and samples. E.T.B., E.W.S. and W.S.H. are NutNet network coordinators. A detailed listing of author contributions can be found in the author contribution matrix (Supplementary Table 6).

## Competing interests

The authors declare no competing interests.

## Additional information

A.C. Risch [1]*, S. Zimmermann [1], R. Ochoa-Hueso [2], M. Schütz[1], B. Frey [1], J.L. Firn [3], P.A. Fay [4], F. Hagedorn[1], E.T. Borer [5], E.W. Seabloom [5], W.S. Harpole[6,7,8], J.M.H. Knops[9,10], R.L. McCulley [11], A.A.D. Broadbent [12,13], C.J. Stevens [13], M.L. Silveira[14], P.B. Adler[15], S. Báez[16], L.A. Biederman [17],

J.M. Blair [18], C.S. Brown [19], M.C. Caldeira[20], S.L. Collins [21], P. Daleo [22], A. di Virgilio[23], A. Ebeling[24], N. Eisenhauer [7,25], E. Esch [26], A. Eskelinen[6,7,27], N. Hagenah[28], Y. Hautier [29], K.P. Kirkman [30], A.S. MacDougall[31], J.L. Moore[32], S.A. Power [33], S.M. Prober[34], C. Roscher [6,7], M. Sankaran[35,36], J. Siebert [7,25], K.L. Speziale [23], P.M. Tognetti [37], R. Virtanen [6,7,27], L. Yahdjian[37] & B. Moser [1]

[1]Swiss Federal Institute for Forest, Snow and Landscape Research WSL, Zuercherstrasse 111, 8903 Birmensdorf, Switzerland. [2]Department of Biology, IVAGRO, University of Cádiz, Campus de Excelencia Internacional Agroalimentario (ceiA3), Campus Rio San Pedro, 11510 Puerto Real, Cádiz, Spain. [3]Queensland University of Technology (QUT), School of Earth, Environmental and Biological Sciences, Science and Engineering Faculty, Brisbane, QLD 4001, Australia. [4]USDA-ARS Grassland Soil, and Water Research Laboratory, Temple, TX 76502, USA. [5]Department of Ecology, Evolution, and Behavior, University of Minnesota, St. Paul, MN, USA. [6]Department of Physiological Diversity, Helmholtz Center for Environmental Research—UFZ, Permoserstrasse 15, Leipzig 04318, Germany. [7]German Centre for Integrative Biodiversity Research (iDiv) Halle-Jena-Leipzig, Deutscher Platz 5e, Leipzig 04103, Germany. [8]Institute of Biology, Martin Luther University Halle-Wittenberg, Am Kirchtor 1, Halle (Saale) 06108, Germany. [9]School of Biological Sciences, University of Nebraska, 211A Manter Hall, Lincoln, NE 68588, USA. [10]Department of Health and Environmental Sciences, Xi'an Jiaotong Liverpool University, Suzhou 215213, China. [11]Department of Plant & Soil Sciences, University of Kentucky, Lexington, KY 40546-0312, USA. [12]School of Earth and Environmental Sciences, Michael Smith Building, The University of Manchester, Oxford Road, Manchester M13 9PT, UK. [13]Lancaster Environment Centre, Lancaster University, Lancaster LA1 4YQ, UK. [14]University of Florida, Range Cattle Research and Education Center, Ona, FL 33865, USA. [15]Department of Wildland Resources and the Ecology Center, Utah State University, 5230 Old Main, Logan, UT 84103, USA. [16]Departamento de Biología, Escuela Politécnica Nacional del Ecuador, Ladrón de Guevera E11-253 y Andalucía, Quito, Ecuador. [17]Department of Ecology, Evolution, and Organismal Biology, Iowa State University, Ames, IA 50011, USA. [18]Division of Biology, Kansas State University, Manhattan, KS 66502, USA. [19]Department of Bioagricultural Sciences and Pest Management, Graduate Degree Program in Ecology, Colorado State University, 1177 Campus Delivery, Fort Collins, CO, USA. [20]Centro de Estudos Florestais, Instituto Superior de Agronomia, Universidade de Lisboa, Tapada da Ajuda, 1349-017 Lisboa, Portugal. [21]Department of Biology, University of New Mexico, Albuquerque, NM 87131, USA. [22]Instituto de Investigaciones Marinas y Costeras (IIMyC), Universidad Nacional de Mar del Plata, CONICET, Mar del Plata, Argentina. [23]INIBIOMA (CONICET-UNCOMA), Universidad Nacional del Comahue, Grupo de Investigaciones en Biología de la Conservación (GrInBiC) Laboratorio Ecotono, Quintral, 1250 Bariloche, Argentina. [24]Institute of Ecology and Evolution, Friedrich-Schiller-University Jena, Dornburger Str. 159, 07743 Jena, Germany. [25]Institute of Biology, Leipzig University, Deutscher Platz 5e, 04103 Leipzig, Germany. [26]University of California San Diego, 9500 Gilman Dr, La Jolla, CA 92037, USA. [27]Department of Ecology and Genetics, University of Oulu, Pentti Kaiteran katu 1, 90014 Oulu, Finland. [28]Mammal Research Institute, Department of Zoology & Entomology, University of Pretoria, Pretoria, South Africa. [29]Ecology and Biodiversity Group, Department of Biology, Utrecht University, Padualaan 8, 3584 CH Utrecht, The Netherlands. [30]University of KwaZulu-Natal, Pietermaritzburg, Private Bag X01, Scottsville 3209, South Africa. [31]Department of Integrative Biology, University of Guelph, Guelph N1G 2W1 ON, Canada. [32]School of Biological Sciences, Monash University, Claytion, VIC 3800, Australia. [33]Hawkesbury Institute for the Environment, Western Sydney University, Locked Bag 1797, Penrith, NSW 2751, Australia. [34]CSIRO Land and Water, Private Bag 5, Wembley, WA 6913, Australia. [35]National Centre for Biological Sciences, TIFR, Bangalore 560065, India. [36]School of Biology, University of Leeds, Leeds LS2 9JT, UK. [37]Universidad de Buenos Aires, Facultad de Agronomía, Instituto de Investigaciones Fisiológicas y Ecológicas vinculadas a la Agricultura (IFEVA), CONICET, Buenos Aires, Argentina. *email: anita.risch@wsl.ch

