## [Peer Review File · Nature Communications]

Reviewers' comments:

Reviewer #1 (Remarks to the Author):

In the manuscript, "Soil net nitrogen mineralization in global grasslands," an experiment is described in which soils were collected from 30 grasslands and analyzed both in-situ, using the resin core method, and in the lab under controlled conditions. The manuscript then uses linear mixed effects models to pull out relationships among variables followed by confirmatory path analysis to provide a potential causal conceptual model to explain variance in rates of net N min in both the field and laboratory assays, as well as allowing relationships between the two response variables. The manuscript suggests that climate variables may drive the measured rates, as well as soil C, soil bulk density, and the abundance of ammonia oxidizing bacteria. The manuscript is in general well written, and the experimental design is fantastic. That being said, there are what I view as two major flaws with the path analysis that undermine the fantastic structure of the experiment, one having to do with the direction of causality and one having to do with the direct or indirect nature of causality.

The first major flaw is that ammonia oxidizing bacteria are included as a purported driver of net N mineralization; the more likely causality instead is that AOB are driven by available ammonium which is captured in the net N min assays as ammonium, nitrite, and nitrate depending on the rapidity and efficiency of the nitrification process. Part of the reasoning for this is that ammonia oxidizers are typically thought to be limited in their activity by substrate availability, namely ammonia on which they rely as an energy source which they use to fix carbon.

While there is an undeniable correlation between AOB and net N mineralization, and while they do drive the first step of the transformation of ammonium to nitrate, they are neither increasing nor decreasing the pool of N in the inorganic nitrogen pool, and therefore they cannot be driving its change in size (net N min) to any appreciable extent. The one exception to this is that they are incorporating a small fraction of N as they produce biomass, though they are typically a small portion of microbial biomass, are generally thought to be slow growing, and this would be a small negative effect.

This error in logic is further compounded in the manuscript by the presentation of this correlation as being similar to patterns linking microbial biomass to net N mineralization. Since microbial biomass captures those organisms serving as the catalysts or potential catalysts of organic matter breakdown and ammonification, the use of microbial biomass is logical and I would have no objection to its use in this analysis. Nor would I have objections to qPCR work attempting to quantify general bacterial and fungal abundance, despite the assumptions necessary to get usable numbers given variability in copy number across bacterial, archaeal, and fungal taxa.

The second major flaw has to do with the application of path analysis to represent direct causal relationships between factors that cannot possibly be directly related to one another, and discussion of them as if they can. For example, I do not understand how one measurement of net N mineralization can influence the other unless it was the same soil being incubated in both cases. From a multiple linear regression or LMM context, sure, they are correlated and so they can be predictive, but I think indicating they are causal is a misapplication of the path analysis technique. Similarly, I do not see how field temperatures can directly influence what is happening in the lab. There is no direct causal relationship when the soils are then incubated at 20 degrees C.

A third but less major concern that I have is that this modeling effort looks at 30 sites and yet the model has a total of eight parameters. Yes, there were a total of 87 samples, but many of those were essential replicates. I think the model may be a bit overly complex for the number of sites included in this analysis. I still think that path analysis is a logical framework if applied differently, but I think it might be useful in the methods to make clear that this an exploratory analysis.

A fourth, but also less major concern is that the paper reads as if it was published before the vast amount of work focusing on the fact that plants successfully compete with microbes for amino acids and other N-containing organic monomers, and that plants also actively compete with microbes for ammonium and nitrate, as described in the cited Schimel and Bennett paper. I would recommend a little more nuance

While I have other more minor comments and concerns, these ones that I have stated are so overwhelming that I cannot in good faith support this manuscript without major reworking of the path analysis, perhaps new sample analyses (e.g., qPCR of 16S and 18S), and thorough revision. My concern is that, with the removal of the AOB and the purported influence of lab net N min on field net N min, the explanatory power of the model may fall apart. Regarding the apparent causal nature of climate factors on lab assays, this can be potentially explained away by looking at it as likely indirect effects not captured by the other variables in the model.

While this review may come across as largely negative, I think the subject matter is important, the experiments seem to be carefully designed and executed, and while I disagree fundamentally on much of the data analysis, I would be excited to see this paper reformulated after some deep work rethinking and redoing the path analysis component. Even if this paper showed that there is an even further limited ability to measure net N min when AOB and direct climate drivers are removed, that would be consistent with other studies and would be a valuable contribution.

Reviewer #2 (Remarks to the Author):

This submission examined the net N mineralization and tested the relationship between lab- and field- incubation methods by using a dataset from 30 grassland (Nutrient Network), and related these results to climate and soil properties. The authors found that temperature, bulk density and soil carbon explained most variation of field net mineralization, and that lab-incubation method could explain variations of field result when combined with soil properties, biotic and climatic parameters. The results are interesting for scientist from Soil science, Agronomy and Ecology. However, given that there have been lots of such studies to related net N mineralization with biotic and abiotic factors and to compare the results from both methods, I do not think this manuscript is novel enough to merit publication in such a high profile multi-discipline journal. I do not think that results from 30 sites could address a global pattern, particularly 13 of 30 were located in North America (Mainly in USA) (Table S1, Fig. S1).

The authors do have done a good job in designing the study and structuring the manuscript. I would recommend the authors submit it to a more specific journal.

Reviewers' comments:

Reviewer #1 (Remarks to the Author):

In the manuscript, "Soil net nitrogen mineralization in global grasslands," an experiment is described in which soils were collected from 30 grasslands and analyzed both in-situ, using the resin core method, and in the lab under controlled conditions. The manuscript then uses linear mixed effects models to pull out relationships among variables followed by confirmatory path analysis to provide a potential causal conceptual model to explain variance in rates of net N min in both the field and laboratory assays, as well as allowing relationships between the two response variables. The manuscript suggests that climate variables may drive the measured rates, as well as soil C, soil bulk density, and the abundance of ammonia oxidizing bacteria. The manuscript is in general well written, and the experimental design is fantastic. That being said, there are what I view as two major flaws with the path analysis that undermine the fantastic structure of the experiment, one having to do

with the direction of causality and one having to do with the direct or indirect nature of causality.

The first major flaw is that ammonia oxidizing bacteria are included as a purported driver of net N mineralization; the more likely causality instead is that AOB are driven by available ammonium which is captured in the net N min assays as ammonium, nitrite, and nitrate depending on the rapidity and efficiency of the nitrification process. Part of the reasoning for this is that ammonia oxidizers are typically thought to be limited in their activity by substrate availability, namely ammonia on which they rely as an energy source which they use to fix carbon.

While there is an undeniable correlation between AOB and net N mineralization, and while they do drive the first step of the transformation of ammonium to nitrate, they are neither increasing nor decreasing the pool of N in the inorganic nitrogen pool, and therefore they cannot be driving its change in size (net N min) to any appreciable extent. The one exception to this is that they are incorporating a small fraction of N as they produce biomass, though they are typically a small portion of microbial biomass, are generally thought to be slow growing, and this would be a small negative effect.

This error in logic is further compounded in the manuscript by the presentation of this correlation as being similar to patterns linking microbial biomass to net N mineralization. Since microbial biomass captures those organisms serving as the catalysts or potential catalysts of organic matter breakdown and ammonification, the use of microbial biomass is logical and I would have no objection to its use in this analysis. Nor would I have objections to qPCR work attempting to quantify general bacterial and fungal abundance, despite the assumptions necessary to get usable numbers given variability in copy number across bacterial, archaeal, and fungal taxa.

Dear reviewer,
many thanks for your kind and supportive comments. We have now done our best to successfully address all your concerns and questions. Please, find our responses to each comment below.

We agree that AOBs are only involved in the nitrification, but ammonification would be completed by all microbes. Therefore, including microbial biomass is essential. Thus, we included this data, which was made available from collaborators working with the same samples. These collaborators were now also added as co-authors. Microbial biomass data were available for all sites, but for two replicates at a site where we had a total of 5 replicates (chilcas.ar). Thus, the dataset now contains 85 observations (instead of 87), but still 30 sites. However, this adjustment also made the data set more balanced.

We decided to keep AOB and AOA abundances within our dataset, given their role in driving a critical step of the mineralisation process. This is also supported by the fact that AOB abundance explains a unique portion of variability in our potential soil net N mineralisation (laboratory measures). Microbial biomass was included in one of the top models for potential soil net N mineralisation, and was important to explain the differences in realised soil net N mineralisation. As microbial biomass was highly correlated with soil organic C ($r = 0.85$), we decided not to retain both variables in the model selection process or the path analysis. However, we conducted our analyses once with microbial biomass and once with soil organic C included. Both approaches yielded very similar findings. We

are now including the models with microbial biomass as a predictor variable in the main text and moved the models including soil organic C into the supplementary material.

Finally, we exchanged bulk density with soil clay content in our SEM as in the new analyses clay content was found to be a stronger predictor of realised soil net N mineralisation compared to bulk density.

We are confident that the changes made address the concerns raised by the reviewer in full and make the manuscript much stronger.

The second major flaw has to do with the application of path analysis to represent direct causal relationships between factors that cannot possibly be directly related to one another, and discussion of them as if they can. For example, I do not understand how one measurement of net N mineralization can influence the other unless it was the same soil being incubated in both cases. From a multiple linear regression or LMM context, sure, they are correlated and so they can be predictive, but I think indicating they are causal is a misapplication of the path analysis technique. Similarly, I do not see how field temperatures can directly influence what is happening in the lab. There is no direct causal relationship when the soils are then incubated at 20 degrees C.

These are important comments that made us realise that a more detailed description of the rationale was needed. Although we understand the concerns raised here, we would like to clarify that we did not want to imply that one measurement of N min influences the other, but that it should be possible to estimate realised net N min when we know the potential of the soil communities to mineralise N in the laboratory. However, we found that potential (lab) and realised (field) soil net N mineralisation (see rationale for changing the wording in comments to the editor) were poorly correlated when looking at their direct relationship. Only when combining climate variables, soil properties and potential soil net N mineralisation, it was possible to estimate realised soil net N mineralisation of our samples. Thus, we show that it would be feasible to estimate realised soil net N min when only information about potential mineralisation and soil/climate properties are available. We carefully adjusted the wording within our manuscript to account for this and clarified our hypotheses, results and discussion to clarify.

The concerns the reviewer raised with regard to use our climatic variables for explaining potential soil net N mineralisation are also understandable. However, we did not see the link between these variables as direct, instantaneous relationships, i.e., that temperature variability or temperature of the wettest quarter affect what is happening in the soil in the laboratory. However, our findings suggest that the two variables stand for a legacy effect of climate in shaping soil properties. The fact that these two variables were more important in explaining potential mineralisation than a specific, individual soil physico-chemical variable in the LMMs indicates this clearly. For this reason, we kept the relationships within our models, but we re-wrote these sections in the revised manuscript to better explain our approach and findings.

We also like to point out for each site and plot, we took the total of 85 paired samples that we used in the field and the laboratory, respectively, within an area of 1 m x 1 m. Thus, the samples incubated in the field and laboratory are highly related to one

another even though they were not exactly the “same” samples. We have made this clear in our methods descriptions.

A third but less major concern that I have is that this modeling effort looks at 30 sites and yet the model has a total of eight parameters. Yes, there were a total of 87 samples, but many of those were essential replicates. I think the model may be a bit overly complex for the number of sites included in this analysis. I still think that path analysis is a logical framework if applied differently, but I think it might be useful in the methods to make clear that this an exploratory analysis.

As the reviewer rightly points out, traditional SEMs are restricted to a minimum number of observations, as there needs to be sufficient degrees of freedom to estimate the whole variance–covariance matrix. However, in our analysis we used a piecewise SEM (see Lefcheck 2016), in which a set of linear structured equations are evaluated individually. This approach allows fitting of smaller data sets and therefore overcomes the limitations of standard SEMs with regard to small samples size (Lefcheck 2016). In addition, this method allows to account for the nested experimental design that we have in our data (Lefcheck 2016). We added these explanations to the method section to assure that readers understand and follow our analyses. In addition, we reduced the complexity of the model to some extent as we deleted distance to the equator from our model, which did not contribute much to the overall model fit.

Lefcheck, J. S. piecewiseSEM: Piecewise structural equation modelling in R for ecology, evolution, and systematics. *Methods Ecol. Evol.* 7, 573–579 (2016).

A fourth, but also less major concern is that the paper reads as if it was published before the vast amount of work focusing on the fact that plants successfully compete with microbes for amino acids and other N-containing organic monomers, and that plants also actively compete with microbes for ammonium and nitrate, as described in the cited Schimel and Bennett paper. I would recommend a little more nuance.

We re-wrote most of our introduction and added that plants and microbes also compete for organic forms of N. We also tried to make clear that we are looking at soil N availability overall and not only plant available N (Schimel and Bennet 2004).

Schimel, J. P. & Bennett, J. Nitrogen mineralization: challenges of a changing paradigm. *Ecology* 85, 591–602 (2004).

While I have other more minor comments and concerns, these ones that I have stated are so overwhelming that I cannot in good faith support this manuscript without major reworking of the path analysis, perhaps new sample analyses (e.g., qPCR of 16S and 18S), and thorough revision. My concern is that, with the removal of the AOB and the purported influence of lab net N min on field net N min, the explanatory power of the model may fall apart. Regarding the apparent causal nature of climate factors on lab assays, this can be potentially explained away by looking at it as likely indirect effects not captured by the other variables in the model.

See our explanations above. We are positive that all the changes made to the manuscript clarified these issues raised by the reviewer and make for a much stronger manuscript.

While this review may come across as largely negative, I think the subject matter is important, the experiments seem to be carefully designed and executed, and while I disagree fundamentally on much of the data analysis, I would be excited to see this paper reformulated after some deep work rethinking and redoing the path analysis component. Even if this paper showed that there is an even further limited ability to measure net N min when AOB and direct climate drivers are removed, that would be consistent with other studies and would be a valuable contribution.

As outlined above, we re-wrote large parts of the manuscript to account for all the points the reviewer raised and added new data as requested, which we think has made the manuscript much stronger. By combining climate and soil variables with potential soil net N min, we managed to estimate realised soil net N min. Consequently, our study should help to gain a better understanding of how soil N availability might be influenced by eutrophication, land or climate change that are threatening grassland ecosystems.

Reviewer #2 (Remarks to the Author):

This submission examined the net N mineralization and tested the relationship between lab- and field- incubation methods by using a dataset from 30 grassland (Nutrient Network), and related these results to climate and soil properties. The authors found that temperature, bulk density and soil carbon explained most variation of field net mineralization, and that lab-incubation method could explain variations of field result when combined with soil properties, biotic and climatic parameters. The results are interesting for scientist from Soil science, Agronomy and Ecology. However, given that there have been lots of such studies to related net N mineralization with biotic and abiotic factors and to compare the results from both methods, I do not think this manuscript is novel enough to merit publication in such a high profile multi-discipline journal. I do not think that results from 30 sites could address a global pattern, particularly 13 of 30 were located in North America (Mainly in USA) (Table S1, Fig. S1).

The authors do have done a good job in designing the study and structuring the manuscript. I would recommend the authors submit it to a more specific journal.

Dear reviewer,
many thanks for your comments. We have now done our best to successfully address all your concerns and questions. Please, find our responses to each comment below.

We agree in that there have been a considerable number of studies that examine how net N mineralisation changes in response to abiotic and biotic conditions. Our study, however, presents a step change in experimental design in relation to past studies for the two following main reasons:

1. Potential and realised soil net N mineralisation are measured simultaneously across 30 globally distributed sites that have very carefully and precisely put into place the same methods to collect the data. We are the first that compare realised and potential soil net N mineralisation at a global scale including six continents. By connecting the two indices and identifying their environmental drivers at the global scale, we provide an important basis to improve future global models of soil N availability. Soil N availability is a crucial driver of plant productivity and soil C-

storage. Our findings also contribute to conceptually advancing our understanding of how global soil N availability may respond to future global anthropogenic influences such as climate change or eutrophication. This is a significant strength and novelty of our study, which we think could be useful and important for a wide readership.

2. With the 30 sites that we have in this experiment, we have captured a wide range of abiotic conditions and types of grasslands across six continents. The site selection in the NutNet protocol asks for grasslands indicative of a region and sites are selected for inclusion by local grassland experts. Consequently, our sites are representative for grasslands worldwide. The reviewer’s comments have helped us to recognize that we had not communicated this clearly enough in the previous version. We now more explicitly demonstrate the range of climatic and edaphic conditions (see the distribution of our grasslands with regard to mean annual precipitation, temperature in Figure 2; ranges of soil edaphic properties within the text and mean values for a series of soil properties per site in Supplementary Table S2; see below). Figure 2 is similar to former Supplementary Figure S1, but now shows the sites by continent, and was moved to the main text (see below).

Figure 2. Geographic and climatic distribution of experimental sites. (A) Location of the 30 NutNet sites where the field experiment was conducted and soil samples were collected for laboratory analyses. (B) Study sites represent a wide range of mean annual temperature (MAT) and mean annual precipitation (MAP) conditions. Our sites also cover a wide range of soil edaphic conditions as described in the main text and shown in Supplementary Table S2.

Supplementary Table S2: Soil edaphic properties at our 30 globally distributed sites on six continents. Site, continent of the site location, soil organic C content (Corg; %), soil total N content (Ntot; %), soil C:N ratio, soil pH, soil sand content (sand; %), soil silt content (Silt; %), soil clay content (Clay; %), water holding capacity (WHC; vol%), and soil bulk density (BD, g cm⁻³). Description of mean annual precipitation and temperature, elevation, grassland type and the coordinates of each site can be found in Supplementary Table S1.

Site	Continent	Corg	Ntot	C:N	pH	Sand	Silt	Clay	WHC	BD
bari.ar	South America	2.3	0.2	14.2	5.6	79.1	17.5	3.4	43.2	0.9
bldr.us	North America	0.9	0.1	11.7	5.7	73.2	15.1	11.8	28.6	1.4
bogong.au	Australia	6.1	0.4	14.7	3.8	71.2	13.2	15.7	49.6	0.8
burrawan.au	Australia	0.9	0.1	16.4	4.7	82.5	12.0	5.5	26.3	1.4

cbgb.us	North America	0.8	0.1	11.1	5.5	88.4	7.3	4.4	25.0	1.1
cdcr.us	North America	2.2	0.1	15.6	5.0	89.9	7.2	3.0	25.9	1.1
cdpt.us	North America	1.1	0.1	11.2	5.6	76.4	13.7	9.9	37.6	1.3
chilcas.ar	South America	4.0	0.4	10.9	5.5	48.2	42.5	9.3	42.1	0.8
comp.pt	Europe	1.2	0.1	13.8	4.4	79.8	15.6	4.6	24.7	1.4
cowi.ca	North America	5.7	0.4	13.0	4.9	58.7	23.6	17.7	33.5	0.6
frue.ch	Europe	3.5	0.4	9.8	4.9	44.8	33.9	21.4	44.5	1.0
jena.de	Europe	5.0	0.5	10.7	6.9	9.1	39.2	51.8	36.6	1.0
kibber.in	Asia	3.3	0.2	21.5	7.6	38.9	36.8	24.3	33.1	1.1
kilp.fi	Europe	7.8	0.6	13.5	3.9	59.8	28.5	11.7	57.0	0.6
koffler.ca	North America	2.6	0.2	11.1	6.9	62.8	27.9	9.4	30.7	1.0
konz.us	North America	3.9	0.3	14.3	5.6	15.6	49.4	35.0	43.2	0.9
lancaster.uk	Europe	22.3	1.3	17.8	4.1	70.6	6.9	22.5	63.8	0.5
marc.ar	South America	4.0	0.4	11.0	7.2	72.1	18.2	9.7	48.7	0.9
mtca.au	Australia	0.8	0.1	15.4	4.4	82.9	10.5	6.6	22.5	1.4
podo.ec	South America	7.5	0.4	19.0	3.3	50.8	36.3	12.9	56.0	0.4
rook.uk	Europe	3.2	0.3	12.3	3.4	83.3	10.7	6.0	41.0	1.1
saline.us	North America	4.1	0.3	15.1	6.7	26.8	44.3	28.9	35.2	1.1
sevi.us	North America	0.3	0.0	9.8	7.7	86.1	8.2	5.7	27.7	1.4
sgs.us	North America	1.1	0.1	10.7	5.1	72.6	15.2	12.2	37.7	1.2
shps.us	North America	2.5	0.2	13.1	7.5	50.5	34.7	14.9	44.7	1.2
spin.us	North America	2.2	0.2	9.1	5.6	14.8	56.7	28.6	43.1	1.1
temple.us	North America	10.0	0.4	23.6	7.3	21.1	25.6	53.3	44.5	0.7
ukul.za	Africa	5.1	0.3	16.1	5.1	12.5	35.8	51.7	39.9	0.9
valm.ch	Europe	4.5	0.3	13.3	4.9	68.0	22.4	9.6	37.7	0.9
yarra.au	Australia	0.9	0.1	11.4	4.5	80.1	15.6	4.3	29.6	1.2

Reviewers' comments:

Reviewer #1 (Remarks to the Author):

In the revised version of "", the authors have gone to great pains to address the specific comments that I provided in response to the original submission and are to be commended. I also thank the authors for referring me to the Lefcheck 2016 paper. That being said, there remain either fundamental misunderstandings of soil N cycling (which I think is unlikely), or misapplication of SEM (more likely), and I still cannot support publication of this manuscript in its current form despite the elegance of the experiment, the nuance of the setup for the SEM, and the novelty of many of the results.

Please excuse a brief digression which I would like to use to set forth my understanding on SEM which I think contrast with those in this manuscript. This is the lens by which I'm seeing fundamental flaws in the model structure put forth in this manuscript. In SEM, the non-directionality of other linear regression techniques is scrapped for a directional and hypothesized causal relationship allowing us to move past "correlation is not causation" to "we hypothesize that this covariance is indicative of this causal relationship.

This may be easiest to see in the diagrams, in which the model represents that the explanatory variable (start of the arrow) is having a direct causal influence on the response variable (head of the arrow) as indicated by a direct connection. Indirect causal influences are represented by another variable falling between the explanatory and response variable of interest. The arrow and equations indicate the direction of this causation, and it is up to the investigator to create a model that incorporates sensible and possible causation. Due to the fact that the underlying math is agnostic with respect to the direction of causality, a model with backwards causation may fit as well (or even better) than one with causation in the appropriate direction.

Turning now to the model at hand, while I do not dispute that AOB may explain unexplained variance in net N min, I assert that the direction of causation is backwards. AOB are not driving N min, but rather, responding to N min. It may improve the model fit, but it represents either a lack of understanding of the role of AOB in N cycling, or a misapplication of SEM. If the goal was to examine the drivers of AOB, then having an arrow that goes the other direction would make perfect sense and I would wholeheartedly be in agreement. As it stands, I view this as a fundamental flaw in the model.

Similarly, I am still uncomfortable with using potential net N min as a causal driver of realized net N min. While they are covary with one another, and while one can be mathematically be used to predict the other, I do not see the logic behind one driving the other. In my thinking, they are two different response variables, and they are responding somewhat similarly to a group of causal drivers. I would be more comfortable if they were shown as being correlated to one another through the use of a double headed arrow, but that gets away from the model structure of trying to use an array of parameters to get at the likely field rates.

Finally, I still object to hypothesizing that the climatic parameters are causal drivers of process rates in a different setting. They are indirect drivers, and the direct arrows between them and the process rates is acceptable, but the language should reflect this still more clearly. I am not asking the authors to make their results equivocal, but rather, to inject more nuance and make it clear that if these missing direct drivers which are influenced by the climate parameters were determined (whether they are protein content, clay mineralogy, or something else), an even stronger causal framework might be developed.

If the analyses were just a GLMM approach, I would not object to the inclusion of AOB or other variables. However, the nature of SEMs is that they are formalizing hypothesized causal relationships. Due to the fact that there are relationships hypothesized in this model that are

counter to established fact, and that others are described where there cannot be a direct causal connection, I cannot support publication of this manuscript in its current form.

Reviewer #2 (Remarks to the Author):

I think the authors had addressed my comments, and the paper is ready to be accepted.

Although we cannot offer to publish your paper in Nature Communications, the work may be appropriate for another journal in the Nature Research portfolio. If you wish to explore suitable journals and transfer your manuscript to a journal of your choice, please use our <https://mts-ncomms.nature.com/cgi-bin/main.plex?el=A3S1BVqR6B7FAgU5X5A9ftdQ2hnDuabUBwqO7F3ZQZ> manuscript transfer portal. If you transfer to Nature-branded journals or to the Communications journals, you will not have to re-supply manuscript metadata and files. This link can only be used once and remains active until used.

All Nature Research journals are editorially independent, and the decision to consider your manuscript will be taken by their own editorial staff. For more information, please see our http://www.nature.com/authors/author_resources/transfer_manuscripts.html?WT.mc_id=EMI_NPG_1511_AUTHORTRANSF&WT.ec_id=AUTHOR manuscript transfer FAQ page.

Reviewer #1 (Remarks to the Author):

In the revised version of “”, the authors have gone to great pains to address the specific comments that I provided in response to the original submission and are to be commended. I also thank the authors for referring me to the Lefcheck 2016 paper. That being said, there remain either fundamental misunderstandings of soil N cycling (which I think is unlikely), or misapplication of SEM (more likely), and I still cannot support publication of this manuscript in its current form despite the elegance of the experiment, the nuance of the setup for the SEM, and the novelty of many of the results.

Please excuse a brief digression which I would like to use to set forth my understanding on SEM which I think contrast with those in this manuscript. This is the lens by which I’m seeing fundamental flaws in the model structure put forth in this manuscript. In SEM, the non-directionality of other linear regression techniques is scrapped for a directional and hypothesized causal relationship allowing us to move past “correlation is not causation” to “we hypothesize that this covariance is indicative of this causal relationship.

This may be easiest to see in the diagrams, in which the model represents that the explanatory variable (start of the arrow) is having a direct causal influence on the response variable (head of the arrow) as indicated by a direct connection. Indirect causal influences are represented by another variable falling between the explanatory and response variable of interest. The arrow and equations indicate the direction of this causation, and it is up to the investigator to create a model that incorporates sensible and possible causation. Due to the fact that the underlying math is agnostic with respect to the direction of causality, a model with backwards causation may fit as well (or even better) than one with causation in the appropriate direction.

-- We agree with the points described by the Reviewer. One of the main advantages of SEMs is to be able to move from correlations towards causal understanding and to lend support for a conceptual model within the limits of a preconceived theoretical framework grounded on the current understanding of the discipline (i.e., data and literature). It is also well known in ecology that interactions between ecological entities and processes are seldomly unidirectional. Thus, if we strictly apply this in SEMs, we would again be left with a mere set of partial correlations. However, very often cause-effect relations occur predominantly in one direction, and thus can be expressed as a single-headed arrow. This also applies for our case, in which *AOB abundance was measured before the field and lab incubations and therefore could not have been affected by end-product accumulation*. Thus, we acknowledge that the reviewer’s assumptions about SEMs are definitely correct and in alignment with our own. To clarify to the reader the reasons for the links and their directionality in our SEM, we added a conceptual figure and a table presenting detailed explanations and supporting citations for our choices (Figure 3, Table 2 in the main text). We also extended the description of the model in the introduction and methods sections. We hope this clarifies the rationale for our conceptual model.

Turning now to the model at hand, while I do not dispute that AOB may explain unexplained variance in net N min, I assert that the direction of causation is backwards. AOB are not driving N min, but rather, responding to N min. It may improve the model fit, but it represents either a lack of understanding of the role of AOB in N cycling, or a misapplication of SEM. If the goal was to examine the drivers of AOB, then having an arrow that goes the other direction would make perfect sense and I would wholeheartedly be in agreement. As it stands, I view this as a fundamental flaw in the model.

-- While we agree with the reviewer that there is evidence supporting AOB responses to N availability (Carney et al. 2004, Hu et al. 2015, Zhang et al. 2015, Pierre et al. 2017), we also

note evidence for the opposite (e.g., Jia and Conrad 2009, Di et al. 2009). Therefore, AOB and N mineralisation are interrelated, which creates what may be a chicken-or-egg scenario; representing two ways of interpreting the AOB and soil N availability relationship, each capturing different aspects of what in an intact natural system functions as a feedback loop. We assessed AOB abundance prior to the incubation. Thus, *in our situation*, the abundance of AOB should drive N process rates and the arrows going from AOB to N mineralisation make sense. As noted above, the point of our SEM model is not to represent all mechanisms involved in the AOB/soil N relationship, but to capture the specific relationships postulated in our study to relate potential to realised N mineralisation. AOB as a driver of N mineralisation was also demonstrated by quite a few other studies (e.g., Di et al. 2009). Please also note that several studies revealed that even when AOB are less abundant than AOA (amoA gene copy numbers), AOB are more closely coupled with measured changes in ammonia oxidation activity (Jia and Conrad 2009, Di et al. 2009, Wakelin et al. 2014, Tao et al. 2017). This is reflected in the findings of Ouyang et al. (2016), where approximately 80-90% of nitrification was attributable to AOB and remainder to AOA. Here, we suggest that AOB are indicative of ammonia oxidation activity (i.e. end product), further supporting the potential soil net N mineralisation associations with AOB populations.

However, to include and acknowledge the reviewers view in the manuscript, we ran the SEM as the reviewer suggested, i.e., with AOB as response variable receiving arrows from potential and realised N mineralisation. We did not find that potential or realised mineralisation had any effect on AOB abundance and the overall model fit became worse. This new analysis as well as an associated explanation was added to the method section as well as the Supplementary Material of the manuscript (Supplementary Fig S8)

Further, we added two additional supplementary figures to the manuscript that show the relationship between AOB abundances, net nitrification and net ammonification (Supplementary Figure S4) as well as the one between microbial biomass and net nitrification and net ammonification (Supplementary Figure S5) for both our laboratory (potential) and field (realised) incubations. As can be seen in these figures, AOB abundance is positively related to potential net nitrification rates and negatively to potential net ammonification (laboratory), but no relationship was found in the field (Supplementary Figure S4). This pattern is exactly what our LMMs as well as arrows in the SEM show. In contrast, higher microbial biomass had a positive effect on both potential and realised net nitrification, but we found no relationship with net ammonification (Supplementary Figure S5).

Similarly, I am still uncomfortable with using potential net N min as a causal driver of realized net N min. While they are covary with one another, and while one can be mathematically be used to predict the other, I do not see the logic behind one driving the other. In my thinking, they are two different response variables, and they are responding somewhat similarly to a group of causal drivers. I would be more comfortable if they were shown as being correlated to one another through the use of a double headed arrow, but that gets away from the model structure of trying to use an array of parameters to get at the likely field rates.

-- The Reviewer raises concerns that we connected potential to realised soil net N mineralisation with a directional arrow in the SEM. The reviewer argues that these two measures are probably correlated and therefore should influence one another and the relationship should be represented by a two headed arrow. We have tested for correlation and our unique dataset derived from coordinated measures of both potential and realised soil net N mineralisation across grasslands globally shows only a weak correlation between potential and realised soil net N mineralisation, as we described in the results section and show in

Supplementary Fig S1). This finding highlights the uncertainty of using potential soil net N mineralisation to predict realised soil net N mineralisation. Consequently, we built our SEM to test whether it would be possible to combine environmental and soil property variables and potential soil net N mineralisation to predict realised soil net N mineralisation, which would be a significant advance in understanding and modelling the N cycle. In this sense we regard potential soil net N mineralisation as a measure for the transformation stage or stability of soil organic nitrogen, which is rather a soil property or status than a process. If we would find (and we did) such an approach to be successful, it might become possible to use potential soil net N mineralisation data widely available in the literature to estimate what might happen under ambient conditions. Such an approach would be highly beneficial for moving our understanding of ecosystem functioning forward. We described our approach thoroughly in the manuscript, but we felt that this was not acknowledged by the reviewer.

Nevertheless, we adjusted our SEM as suggested by the reviewer (see Figure below) and redid our calculations. As can be seen in the Figure below, the prediction of realised soil net N mineralisation is much poorer when we consider a correlation structure between potential and realised N mineralisation compared to our original. Hence, we decided to keep our initial model within the revised manuscript. For clarity reasons we did not include this new model in the Supplement, however, if the editor thinks that we should include it, we certainly can.

Finally, I still object to hypothesizing that the climatic parameters are causal drivers of

process rates in a different setting. They are indirect drivers, and the direct arrows between them and the process rates is acceptable, but the language should reflect this still more clearly. I am not asking the authors to make their results equivocal, but rather, to inject more nuance and make it clear that if these missing direct drivers which are influenced by the climate parameters were determined (whether they are protein content, clay mineralogy, or something else), an even stronger causal framework might be developed.

-- We added more nuanced language to the manuscript and now also specifically mention that our links between climate and process rates are included in the SEM as they stand for 1) long-term climate influencing soil properties in general, without being able to detect which ones specifically, or 2) our numerous analyses of different soil properties did not include the variables that would have exerted the direct effects of soil properties on process rates across global grasslands.

If the analyses were just a GLMM approach, I would not object to the inclusion of AOB or other variables. However, the nature of SEMs is that they are formalizing hypothesized causal relationships. Due to the fact that there are relationships hypothesized in this model that are counter to established fact, and that others are described where there cannot be a direct causal connection, I cannot support publication of this manuscript in its current form.

-- We respectfully but strongly disagree that the relationships in our SEM are counter to established facts or cannot be direct causal connections. We have supported our argumentation by providing a detailed listing of all the links included in the SEM (Fig 3, Table 2) and literature that shows that this approach is justified. Further, *our SEM represents a novel "working model" for how we might be able to use data on potential soil net N mineralisation to predict the rates in the field.* This is a step forward from describing potential soil net N mineralisation as has already been done with great success by other authors (e.g., Colman and Schimel 2013, Liu et al. 2016, 2017). We believe our study adds considerable new understanding of factors affecting soil net N mineralisation in grassland ecosystems.

References

- Carney, K. M. et al. 2004. Diversity and composition of tropical soil nitrifiers across a plant diversity gradient and among land-use types. - *Ecol. Lett.* 7: 684–694.
- Colman, B. P. and Schimel, J. P. 2013. Drivers of microbial respiration and net N mineralization at the continental scale. - *Soil Biol. Biochem.* 60: 65–76.
- Di, H. J. et al. 2009. Nitrification driven by bacteria and not archaea in nitrogen-rich grassland soils. - *Nat. Geosci.* 2: 621.
- Hu, H. et al. 2015. The large-scale distribution of ammonia oxidizers in paddy soils is driven by soil pH, geographic distance, and climatic factors. - *Front. Microbiol.* 6: 938.
- Jia, Z. and Conrad, R. 2009. Bacteria rather than archaea dominate microbial ammonia oxidation in an agricultural soil. - *Environ. Microbiol.* 11: 1658–1671.
- Liu, Y. et al. 2016. Patterns and regulating mechanisms of soil nitrogen mineralization and temperature sensitivity in Chinese terrestrial ecosystems. - *Agric. Ecosyst. Environ.* 215: 40–46.
- Liu, Y. et al. 2017. A global synthesis of the rate and temperature sensitivity of soil nitrogen mineralization: Latitudinal patterns and mechanisms. - *Glob. Chang. Biol.* 23: 455–464.
- Ouyang, Y. et al. 2016. Ammonia-oxidizing bacteria are more responsive than archaea to nitrogen source in an agricultural soil. - *Soil Biol. Biochem.* 96: 4–15.
- Pierre, S. et al. 2017. Ammonia oxidizer populations vary with nitrogen cycling across a tropical montane mean annual temperature gradient. - *Ecology* 98: 1896–1907.
- Tao, R. et al. 2017. Response of ammonia-oxidizing archaea and bacteria in calcareous soil to mineral and organic fertilizer application and their relative contribution to nitrification. -

Soil Biol. Biochem. 114: 20–30.

Wakelin, S. et al. 2014. Predicting the efficacy of the nitrification inhibitor dicyandiamide in pastoral soils. - *Plant Soil* 381: 35–43.

Zhang, J. et al. 2015. Distribution of ammonia-oxidizing archaea and bacteria in plateau soils across different land use types. - *Appl. Microbiol. Biotechnol.* 99: 6899–6909.

Reviewers' comments:

Reviewer #1 (Remarks to the Author):

I wholeheartedly agree with the authors that AOB drive nitrification. However, this model is a model about net N mineralization, not nitrification. As such, my concerns about using AOB in the model as a driver of net N mineralization (potential or realized) remain consistent from my first and second reviews of this manuscript. I will elaborate on why and address a few other points that I raised and to which the authors responded.

I'll start my critique on the inclusion of AOB as a causal driver of net N mineralization with my perspective (sorry) on methods for measuring N cycling. When one measures gross rates of N mineralization, one does not even look at nitrate, just ammonium. On the timescales over which net N mineralization is measured, much of the N turned into ammonium is converted to nitrate by AOB. As such, to understand the net rates of mineralization, one looks at the accumulation of both ammonium and nitrate to look at the entirety of the mineralized N pool. An important thing to note is that nitrification by AOB does not change the size of the mineralized N pool (except for the tiny fraction they incorporate to slowly build biomass). Instead, AOB change the distribution among forms found within that mineralized pool. While AOB drive the conversion of ammonium to nitrate, they do not drive the rate of net N mineralization.

There is not a mechanistic basis that I am aware of which one can use to rationalize that AOB could be driving the accumulation of inorganic nitrogen in soil. It is neither a chicken and egg paradox, nor is it a feedback loop. In both Jia and Conrad 2009 and Di et al. 2009, it is demonstrated that with the addition of exogenous ammonium, AOB abundance and activity both will respond by increasing due to the increased substrate availability. These papers were cited in the rebuttal as evidence of the role of AOB in driving net N mineralization; however, these papers do not show this. Instead, they show that AOB are important in driving the distribution of N among the two different mineral forms, but not that they increase the total pool size of DIN. Since AOB are not mineralizing nitrogen, they cannot be driving the rate of accumulation of mineral forms of nitrogen. A much more parsimonious explanation is that soils with high numbers of AOB, they serve as an index of past N availability. As such, I cannot endorse drawing an arrow in an SEM from AOB to net N mineralization to represent a causal mechanism that does not exist.

I would recommend that the authors remove AOB from the model completely. I had suggested that drawing arrows from net N min to AOB would be more logical and consistent with causation, but the model is a model of net N mineralization so I don't think it is necessarily useful. Also, the authors raise a good point that the AOB were measured prior to net N mineralization. Though AOB do explain unexplained variance in the model, the authors have not made a clear case for how there could be a causal linkage by which AOB could possibly drive net N mineralization after three versions of the manuscript and two rebuttals. Net N nitrification, yes, but this is not a model of net N nitrification. I recommend removing AOB from the model.

I appreciate the inclusion of S4 and S5. I am encouraged by the correlation between net nitrification and AOB, as it is consistent with other studies. I would argue that the negative

correlation between net ammonification and AOB is as well, as it is consistent with AOB converting ammonium to nitrate.

One interesting option that the authors do have at their disposal would be to put together a model of net nitrification and including AOB in that model (and possibly pH and other factors). I don't think that would fit in the main body of the manuscript but it could make for an interesting exercise and supplemental figure. It also would give an outlet for the AOB data.

With regards to the nature of the relationship between lab and field incubations, after rereading and reflecting on it, I think the authors have a good point that a directional relationship could be acceptable here. I still disagree that it is the most appropriate path, but it is an acceptable one given the way that it is presented. I apologize for missing that distinction in my previous reading of the manuscript.

In summary, with the exception of the assignment of a causal linkage between AOB and net N mineralization, the authors have addressed my concerns with this manuscript. I greatly respect the time and effort the authors put into respectfully disagreeing with me on so many points, and appreciate the work that they have done to alter the manuscript in an effort to both appease the reviewer(s) and also improve the manuscript. I do maintain that having AOB as causal drivers of net N mineralization is contrary to the available evidence, including that presented by the authors. If AOB were removed from the model and if the model fit and explanatory power was still reasonable, I would be supportive of the publication of this manuscript. If that causal linkage is maintained, in conflict with the available evidence, I would not be able to support the publication of this manuscript.

Reviewer #1 (Remarks to the Author):

Dear Dr Risch,

Your manuscript entitled "Soil net nitrogen mineralisation across global grasslands" has now been seen by 1 referees, whose comments are appended below. You will see from their comments copied below that the referees acknowledge the improvements of your revised work, but between them, they also raise a number of remaining concerns, which must prevent us from offering to publish the paper in its present form. It is not clear to us whether you will be able to address all the concerns raised. If you would like to pursue publication in Nature Communications, we will therefore need to see your responses to the criticisms raised and to some editorial concerns, along with a revised manuscript, before we can reach a decision regarding publication.

The referees' reports seem to be quite clear. Naturally, we will need you to address all of the points raised. Specifically, for publication in Nature Communications to be appropriate, we will need you to either provide convincing evidence on the selected AOB or remove it completely from the model (Reviewer #1).

Many thanks to the editor and the reviewer for the opportunity to revise the manuscript and the continued constructive comments. We have provided detailed responses to all the points raised below in blue. In particular, we removed AOB from the SEM, as requested by reviewer #1. These changes made it necessary to adjust the text as well as add, change and remove some of the Supplementary figures to allow the reader to follow the analyses. We believe that these changes made based on the reviewer's comments have further improved our manuscript and we are confident that it has now reached the requested standard for publication in Nature Communications.

The editorial policy checklist, the reporting summary as well as a source file were uploaded to the electronic submission system. The data availability statement is included in the manuscript.

Reviewers' comments:

Reviewer #1 (Remarks to the Author):

I wholeheartedly agree with the authors that AOB drive nitrification. However, this model is a model about net N mineralization, not nitrification. As such, my concerns about using AOB in the model as a driver of net N mineralization (potential or realized) remain consistent from my first and second reviews of this manuscript. I will elaborate on why and address a few other points that I raised and to which the authors responded.

I'll start my critique on the inclusion of AOB as a causal driver of net N mineralization with my perspective (sorry) on methods for measuring N cycling. When one measures gross rates of N mineralization, one does not even look at nitrate, just ammonium. On the timescales over which net N mineralization is measured, much of the N turned into ammonium is converted to nitrate by AOB. As such, to understand the net rates of mineralization, one looks at the accumulation of both ammonium and nitrate to look at the entirety of the mineralized N pool. An important thing to note is that nitrification by AOB does not change the size of the mineralized N pool (except for the tiny fraction they incorporate to slowly build biomass).

Instead, AOB change the distribution among forms found within that mineralized pool. While AOB drive the conversion of ammonium to nitrate, they do not drive the rate of net N mineralization.

There is not a mechanistic basis that I am aware of which one can use to rationalize that AOB could be driving the accumulation of inorganic nitrogen in soil. It is neither a chicken and egg paradox, nor is it a feedback loop. In both Jia and Conrad 2009 and Di et al. 2009, it is demonstrated that with the addition of exogenous ammonium, AOB abundance and activity both will respond by increasing due to the increased substrate availability. These papers were cited in the rebuttal as evidence of the role of AOB in driving net N mineralization; however, these papers do not show this. Instead, they show that AOB are important in driving the distribution of N among the two different mineral forms, but not that they increase the total pool size of DIN. Since AOB are not mineralizing nitrogen, they cannot be driving the rate of accumulation of mineral forms of nitrogen. A much more parsimonious explanation is that soils with high numbers of AOB, they serve as an index of past N availability. As such, I cannot endorse drawing an arrow in an SEM from AOB to net N mineralization to represent a causal mechanism that does not exist.

We very much appreciate the reviewer taking the time to elaborate on the AOB discussion in such detail. These comments have clarified our understanding of the reviewer's concerns and we have revised our manuscript accordingly. This has greatly improved the quality of our manuscript.

I would recommend that the authors remove AOB from the model completely. I had suggested that drawing arrows from net N min to AOB would be more logical and consistent with causation, but the model is a model of net N mineralization so I don't think it is necessarily useful. Also, the authors raise a good point that the AOB were measured prior to net N mineralization. Though AOB do explain unexplained variance in the model, the authors have not made a clear case for how there could be a causal linkage by which AOB could possibly drive net N mineralization after three versions of the manuscript and two rebuttals. Net N nitrification, yes, but this is not a model of net N nitrification. I recommend removing AOB from the model.

We have removed AOB from the SEM, as suggested, and the model still has a very good fit. It explains 33% of the variation in realised soil net N mineralisation across our global grasslands (marginal R^2) opposed to 34% in the SEM with AOB included. We have also deleted Supplementary Figure S8 in which we had reversed the arrows from potential and realised soil net N mineralization to AOB as suggested. We have adjusted the text and Table 1 where we provide the rationale for the choice of pathways included in the model. Figure 3 has been adjusted to show the correct conceptual model, and we exchanged Figure 5A & B as well as Fig S6A & B with versions that contain the results of the new analyses without AOB included in the model. We also adjusted the text to explain that although AOB abundance does explain some of the variation in potential soil net N mineralization in the LMM, including AOB in the SEM has no mechanistic basis from which we could hypothesize that AOB is driving net N mineralisation.

I appreciate the inclusion of S4 and S5. I am encouraged by the correlation between net nitrification and AOB, as it is consistent with other studies. I would argue that the negative correlation between net ammonification and AOB is as well, as it is consistent with AOB converting ammonium to nitrate.

Thank you. We are glad that these figures were meaningful.

One interesting option that the authors do have at their disposal would be to put together a model of net nitrification and including AOB in that model (and possibly pH and other factors). I don't think that would fit in the main body of the manuscript but it could make for an interesting exercise and supplemental figure. It also would give an outlet for the AOB data.

We thank the reviewer for this interesting suggestion. We have fitted this model and included it into the Supplementary material as suggested (now Supplementary Fig S8). We also added rationale for why we constructed this SEM in the methods section.

With regards to the nature of the relationship between lab and field incubations, after rereading and reflecting on it, I think the authors have a good point that a directional relationship could be acceptable here. I still disagree that it is the most appropriate path, but it is an acceptable one given the way that it is presented. I apologize for missing that distinction in my previous reading of the manuscript.

Thank you for re-reading these sections. We are grateful for the reviewer's input.

In summary, with the exception of the assignment of a causal linkage between AOB and net N mineralization, the authors have addressed my concerns with this manuscript. I greatly respect the time and effort the authors put into respectfully disagreeing with me on so many points, and appreciate the work that they have done to alter the manuscript in an effort to both appease the reviewer(s) and also improve the manuscript. I do maintain that having AOB as causal drivers of net N mineralization is contrary to the available evidence, including that presented by the authors. If AOB were removed from the model and if the model fit and explanatory power was still reasonable, I would be supportive of the publication of this manuscript. If that causal linkage is maintained, in conflict with the available evidence, I would not be able to support the publication of this manuscript.

We are grateful to reviewer #1 for the insightful and constructive feedback throughout the reviewing process, which has substantially improved the quality of the manuscript. We genuinely appreciate all the time and effort the reviewer put into commenting on our manuscript, the patience with us as we tried to defend our initial approach, and for discussing several central points of this manuscript with us in such detail. We removed the AOB from the SEM. The fits are still very similar to the old model with very similar explanatory power. We hope that the reviewer agrees that the changes resolve the concerns expressed and that this version of the manuscript for publication can now be supported.

REVIEWERS' COMMENTS:

Reviewer #1 (Remarks to the Author):

I very much appreciate that the authors of this manuscript have taken the time to respond so clearly and thoroughly to the concerns I raised regarding AOB and net N mineralization. With the incorporation of those changes, the manuscript is now at a point where I can support it for publication pending some minor revisions that may have slipped by amidst the major revisions.

There are two lingering issues I see, the first is in lines 239 to 241, and the second is in lines 246 to 251. In lines 239 to 241, the text still states that AOB abundance might be driving net N mineralization due to its effects on nitrification. Given that nitrification does not do anything but change one inorganic form to another, then this cannot be a driver of the rate of accumulation of inorganic nitrogen. Given that the authors have removed that connection in the model, and have agreed in their response with the reasoning that AOB cannot drive net N min, I am assuming that this was accidentally left in the text and Table 2. I would strongly recommend removing this language from the text. While it can be used in table 2, the interpretation that AOB might be driving net N min is very tempting, and so the authors might be better off removing it, though I realize that would entail rerunning the models and remaking the table. While the authors could simply discuss the fact that AOB were correlated, and as such, they ended up in the list of potential drivers in some of their models of potential net N min, given that these models were used in part to inform the SEM efforts I would recommend removing them altogether from these models.

As for the text in lines 246 to 251, here it is implied that since higher microbial biomass correlates with higher net nitrification and since microbial biomass was identified as a driver of net N min, there is an effect of AOB abundance on net N min. Again, there is a correlation, but there is not causation here, nor can there be. While I am a fan of unpacking the "black box" of the soil microbial community, I would argue that this is not evidence that it is necessary for understanding net N mineralization. I would recommend reworking this part of the paragraph and removing the bit about unpacking the black box.

Minor suggestions:

Line 149: I would suggest replacing "soil Nmin" with soil net Nmin

Lines 151-153: I might ditch the letter delimiters of the sublist within the list, as I don't know that they help clarify

Line 217: Technically, through the inclusion of tundra/boreal sites, the study might be outside of just one biome. A minor point, but might want to change this to "a single vegetation type"

Line 264: Consider changing "climate leads" to "climate may lead"

Consider changing "net N nitrification" (which I used in my review for some daft reason, and which may be why it is used here) to "net nitrification" which is a much more common way to refer to the measurement.

REVIEWERS' COMMENTS:

Reviewer #1 (Remarks to the Author):

I very much appreciate that the authors of this manuscript have taken the time to respond so clearly and thoroughly to the concerns I raised regarding AOB and net N mineralization. With the incorporation of those changes, the manuscript is now at a point where I can support it for publication pending some minor revisions that may have slipped by amidst the major revisions. There are two lingering issues I see, the first is in lines 239 to 241, and the second is in lines 246 to 251.

We sincerely thank you for all the important questions raised during the entire review process, your patient and detailed explanations of your concerns, and for taking the time to go carefully through all the changes included in the latest version in detail.

In lines 239 to 241, the text still states that AOB abundance might be driving net N mineralization due to its effects on nitrification. Given that nitrification does not do anything but change one inorganic form to another, then this cannot be a driver of the rate of accumulation of inorganic nitrogen. Given that the authors have removed that connection in the model, and have agreed in their response with the reasoning that AOB cannot drive net N min, I am assuming that this was accidentally left in the text and Table 2. I would strongly recommend removing this language from the text. While it can be used in table 2, the interpretation that AOB might be driving net N min is very tempting, and so the authors

might be better off removing it, though I realize that would entail rerunning the models and remaking the table. While the authors could simply discuss the fact that AOB were correlated, and as such, they ended up in the list of potential drivers in some of their models of potential net N min, given that these models were used in part to inform the SEM efforts I would recommend removing them altogether from these models.

Thank you for helping to nail this important aspect of our manuscript. In this revised version, we adjusted the text to clearly state that AOB was not driving potential soil net N min. The passage now reads as following:

“Potential soil net N_{\min} was higher when more AOB were present at the start of the incubation (Model 5, 6, Table 2). However, there is no mechanistic link between AOB and potential soil net N_{\min} , because AOB only transform ammonium to nitrate, but do not drive net production of total inorganic nitrogen in the soil. Yet, AOB abundance was positively correlated with potential net nitrification (Supplementary Figure 4), which is similar to previous findings^{36–39}. ”

With these changes we were able to retain Supplementary Fig S4 and S5 as well as the SEM for nitrification (Supplementary Fig S8) that demonstrates the linkages between microbial community composition and soil functioning.

As for the text in lines 246 to 251, here it is implied that since higher microbial biomass correlates with higher net nitrification and since microbial biomass was identified as a driver of net N min, there is an effect of AOB abundance on net N min. Again, there is a correlation, but there is not causation here, nor can there be. While I am a fan of unpacking the “black box” of the soil microbial community, I would argue that this is not evidence that it is necessary for understanding net N mineralization. I would recommend reworking this part of the paragraph and removing the bit about unpacking the black box.

Apologies, we did not want to imply that AOB drives net N min, this slipped past our radar in the last round of revisions. We have now corrected the text and also removed the section about the “black box”. The passage now reads:

“However, higher potential soil net N_{\min} was also explained by higher soil microbial biomass alone (Model 7; Table 2) and microbial biomass was positively correlated with potential soil net nitrification (Supplementary Figure 5). Our results agree with findings of a recent meta-analysis that identified soil microbial biomass as an important driver of potential soil net N_{\min} ¹⁹. In addition, the effect of microbial biomass on potential soil net N_{\min} indicates that a quantitatively improved understanding of the soil microbial community could likely improve soil biogeochemical models^{40,41}. ”

Minor suggestions:

Line 149: I would suggest replacing “soil Nmin” with soil net Nmin
Changed as suggested

Lines 151-153: I might ditch the letter delimiters of the sublist within the list, as I don’t know that they help clarify
We deleted the letter delimiters of the sub-list as suggested.

Line 217: Technically, through the inclusion of tundra/boreal sites, the study might be outside of just one biome. A minor point, but might want to change this to “a single

vegetation type”

Changed as suggested

Line 264: Consider changing “climate leads” to “climate may lead”

Changed as suggested

Consider changing “net N nitrification” (which I used in my review for some daft reason, and which may be why it is used here) to “net nitrification” which is a much more common way to refer to the measurement.

We changed “net N nitrification” to “net nitrification” throughout the manuscript, adjusted Supplementary Fig S4, S5, and S8 in the same way. We also changed “net N ammonification” to “net ammonification” in Supplementary Fig S4 and S5.